# Megaripple mechanics: bimodal transport ingrained in bimodal sands

Katharina Tholen[1], Thomas Pähtz [2✉], Hezi Yizhaq[3], Itzhak Katra[4✉] & Klaus Kroy [1✉]

Aeolian sand transport is a major process shaping landscapes on Earth and on diverse celestial bodies. Conditions favoring bimodal sand transport, with fine-grain saltation driving coarse-grain reptation, give rise to the evolution of megaripples with a characteristic bimodal sand composition. Here, we derive a unified phase diagram for this special aeolian process and the ensuing nonequilibrium megaripple morphodynamics by means of a conceptually simple quantitative model, grounded in the grain-scale physics. We establish a well-preserved quantitative signature of bimodal aeolian transport in the otherwise highly variable grain size distributions, namely, the log-scale width (Krumbein phi scale) of their coarse-grain peaks. A comprehensive collection of terrestrial and extraterrestrial data, covering a wide range of geographical sources and environmental conditions, supports the accuracy and robustness of this unexpected theoretical finding. It could help to resolve ambiguities in the classification of terrestrial and extraterrestrial sedimentary bedforms.

[1] Institute for Theoretical Physics, Leipzig University, Leipzig, Germany. [2] Institute of Port, Coastal and Offshore Engineering, Ocean College, Zhejiang University, 866 Yu Hang Tang Road, 310058 Hangzhou, China. [3] Department of Solar Energy and Environmental Physics, Blaustein Institutes for Desert Research, Ben-Gurion University of the Negev, Sede Boqer Campus, Be'er Sheva, Israel. [4] Department of Geography and Environmental Development, Ben-Gurion University of the Negev, Be'er Sheva, Israel. ✉email: 0012136@zju.edu.cn; katra@bgu.ac.il; klaus.kroy@uni-leipzig.de

f exposed to atmospheric flows, planetary surfaces composed of loose sand may continuously evolve into dynamic landscapes. The most common aeolian bedforms found on Earth are decimeter-sized ripples and dunes ranging from tens to hundreds of meters in size[1,2]. They are generally composed of a surprisingly well-defined selection of fine sands[3,4] (here meaning sediments of grains larger than about 60 μm), technically characterized by their narrowly peaked unimodal grain-size distribution (GSD). A more perplexing third type of bedform with sand waves of intermediate size (30 cm to several meters in wavelength on Earth[5,6]), between ripples and dunes, is commonly referred to as megaripple[5,7], gravel or pebble ridge[3], or granule, giant or pebble ripple[8,9]. Megaripples have more gently sloped cross-sectional profiles[10,11] and less regular crest-lines, spacings and alignments than the smaller ripples[5,8,10,12–15] (Fig. 1 and Supplementary Fig. S9). Another unique trait is their less uniform GSD, which exhibits a characteristic bimodality (Fig. 1a). In particular, they feature a conspicuous coarse-grain fraction, most abundant on the windward slopes near the megaripple crests, which prompts the suggestive notion of an "armouring layer"[4,6,8,11–13,16–19]. Various laboratory and field studies have characterized the bimodal GSDs statistically by their overall mean grain size or cumulative percentile values, or by more complex approaches[16]. The aim was to thereby relate the GSDs to megaripple morphology (e.g., size, wavelength, transverse shape) and age[6,20], or to discriminate megaripples from other bedforms[21]. The degree of grain-size segregation was also related to the sensitivity of megaripples to variations in wind strength[22–26]. In fact, while megaripples grow quite slowly, they can relatively quickly be destroyed by gusts that exceed the prevailing wind strength, which is yet another anomalous feature that sets them apart from the less fragile ripples and dunes[3,27–30].

Since Bagnold's early observations[3] and the pioneering work of Anderson and Bunas[31], various models have been proposed to explain the formation of megaripples[21,32–39]. They all share the key hypothesis that megaripples originate from a bimodal sand

transport mechanism, namely by fine grains kicking coarse grains (Fig. 1b). The fine grains are accelerated by the wind into a bouncing or *saltating* motion, while the coarse grains only advance incrementally, by a creeping motion known as *reptation*, upon impact. While details are still under investigation, it is generally thought that the instability leading to the emergence of megaripples from a flat bed involves spatial variations in the reptation flux. So far, most models take the presence of bidisperse sand for granted to predict its spatial sorting, with coarse grains accumulating near the crest and fine grains in the troughs, due to their dissimilar travel distances[21,31–33,36,37] or immobilization rates[33,37]. However, more recent theoretical work revealed that bimodal GSDs are not a precondition of megaripple formation but rather co-evolve with the grain-size-dependent aeolian transport and structure formation[38]. Bimodal surface GSDs gradually emerge from a unimodal bulk GSD in undersaturated wind conditions. The ensuing winnowing of fine grains increasingly diminishes their relative concentration in the surface GSD and the resulting formation a non-erodible armouring layer in turn promotes undersaturated fine-grain saltation. This interplay between sand sorting and morphological evolution naturally explains the slow coarsening and fast destruction of megaripples and their armouring layers[23–26]. However, despite the widely appreciated qualitative correlation between megaripple evolution and aeolian transport regimes[3,22–26,38,40], a more precise physical characterization of the role of the environmental conditions has remained elusive. The latter would require quantitative modeling of the threshold wind strengths separating the diverse transport regimes as functions of the grain size and ambient conditions. But these thresholds have proven difficult to measure and unambiguously define even for monodisperse sand[41]. Intuitively, grain polydispersity should aggravate the challenge[42–44] as grain sorting and bimodal transport may feedback onto the thresholds. Moreover, the emerging shapes of the surface GSDs are quite volatile and contingent on the recent wind conditions (Fig. 1a)—apparently without persistent quantitative signatures.

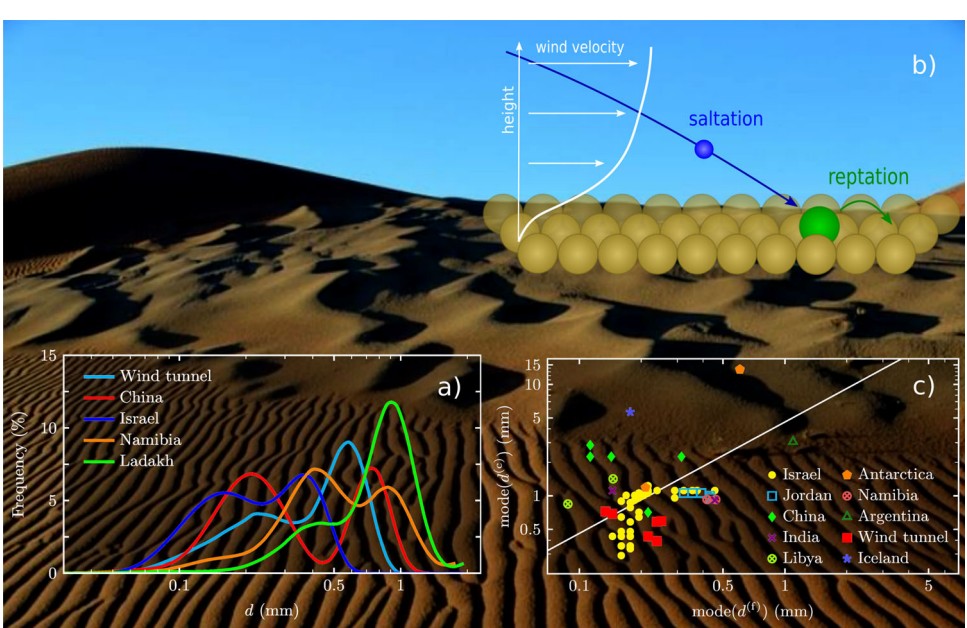

**Fig. 1 Bimodal grain transport creates delicate intermediate-sized megaripples via sand sorting.** *Background photo* (Sossusvlei, Namibia): dunes (top), megaripples (middle) and ripples (bottom) emerge due to aeolian (wind-driven) sand transport. **a** Grain-size distributions (GSDs) found on megaripple crests exhibit a characteristic, yet variable bimodal structure; GSDs are from Nahal Kasuy, Israel[22]; Sanshan Desert, China[16]; Sossusvlei, Namibia (Supplementary Fig. S7); Ladakh, India (Fig. S8) and wind tunnel[20]. **b** Fine-grain saltation drives coarse-grain reptation. **c** Grain-size modes (locations of peak maxima), mode($d^{(f)}$) and mode($d^{(c)}$), in the bimodal crest GSDs appear to be poorly correlated even within the range of terrestrial conditions; data are from refs. [3,6,13,15,16,20–22,82,84,85] (Table S3), with mean ratio mode($d^{(c)}$)/mode($d^{(f)}$) ≈ 4.59 (solid line, coefficient of determination $R^2 = 0.13$).

Even Bagnold's often quoted rule[3,5,6,13,15,18,26,33,34,37,40] that the characteristic coarse-grain size is more than 6 times the characteristic fine-grain size is not borne out by a comprehensive survey of the available literature data (Fig. 1c).

Here, we develop a quantitative grain-scale theory to uncover a robust and precise connection between the coevolving bedforms and bimodal GSDs. The theory does not rely on the specific nature of the instability mechanism causing megaripple formation. It requires only two well-established generic ingredients:

(i)   the fine-grain saltation flux is strongly undersaturated and therefore does not significantly disturb the wind speed, and
(ii)  megaripples and their surface GSDs co-evolve through winnowing of saltating fine grains and accumulation of exclusively reptating coarse grains.

The theory accounts for the diverse nonequilibrium dynamical modes of megaripple evolution and sand sorting under arbitrary aeolian transport conditions. By idealizing the GSD as a mixture of two sharply defined grain populations, we can thereby construct a phase diagram of aeolian transport modes, including the precise shapes of the phase boundaries (Results). The theory also predicts that bimodal transport gives rise to a robust quantitative signature in the otherwise quite volatile continuous megaripple GSDs measured in the field (Fig. 1a), namely, a fixed width of the coarse-grain peak over a logarithmic axis, henceforth called *log-scale width* ($\propto$ Krumbein phi scale width). This dimensionless quantity is the logarithm of the size ratio of the coarsest grains participating in reptation and saltation, respectively, which we herein refer to as *max-size ratio*. For terrestrial conditions, its precise value can be gleaned from the quantitative transport phase diagram (Figs. 2, 3), as suggestively illustrated in Fig. S1. Furthermore, the theory yields an analytical scaling function for the max-size ratio in terms of transport parameters and predicts an unexpected data collapse (Fig. 4). An extensive data compilation encompassing a wide range of geographical sources and environmental conditions on Earth and Mars corroborates our predictions (Fig. 5).

## Results

**Phase diagram for bidisperse sand**. Megaripple morphology and grain sorting are intimately linked. To establish their formal relation, summarized in Table 1, it is useful to consider an idealized GSD of bimodal sands, consisting of only two types of grains with diameters $d^{(f)}$ and $d^{(c)}$, respectively [3,22–26,38,40]. Such bidisperse sand is inert (*no transport*) until the wind shear stress $\tau$ exceeds the threshold value $\tau_t(d^{(f)})$ required to sustain saltation of the fine-grain fraction. For larger $\tau$, up to the saltation threshold $\tau_t(d^{(c)})$ of the coarse-grain fraction (defined shortly), only fine grains saltate. Upon bed impact, they can supply the coarse grains with enough energy to excite them into a small reptation move (*bimodal transport*) or not (*selective transport*)—depending on whether the wind strength $\tau$ exceeds the coarse-grain reptation threshold[38] $\tau_r(d^{(c)})$, which is itself a function of the prescribed

grain-size ratio $d^{(c)}/d^{(f)}$. Finally, for $\tau > \tau_t(d^{(c)})$, all grains saltate, resulting in a more indiscriminate, *unimodal transport* mode.

If we accept the notion of megaripples as bedforms that self-assemble from the reptating coarse grains, the above transport regimes map onto corresponding morphodynamic regimes of megaripple evolution, as indicated in Table 1. Likewise, corresponding sorting regimes can be delineated. Importantly, wind conditions that allow solely the fine grains to saltate, i.e., $\tau_t(d^{(f)}) < \tau < \tau_t(d^{(c)})$, promote fine-grain erosion and the formation of an armouring layer. As a result, the bimodal transport regime exhibits the richest feedback between discriminative transport, sorting, and morphodynamics. Their mutual reinforcement fosters megaripple evolution.

The above qualitative distinction between the various thresholds is not original but can be found in several previous studies[3,22–25,38,40]. However, while only rough empirical estimates for them were offered

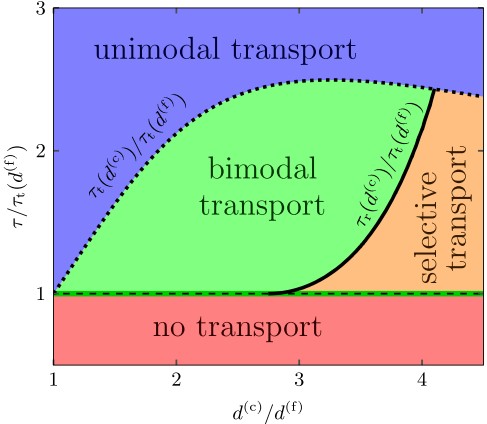

**Fig. 2 Aeolian transport phase diagram for perfectly bidisperse sand (terrestrial conditions).** Here, the sand bed is idealized as consisting of two grain species with diameters $d^{(f)}$ and $d^{(c)}$. The coarse grains can be incrementally kicked forward by the fine grains, resulting in a creeping motion known as reptation, if the wind shear stress $\tau$ falls within the bimodal transport regime (green-shaded area). It is delimited by the saltation thresholds $\tau_t(d^{(f)})$ and $\tau_t(d^{(c)})$ (dashed and dotted lines) of the fine and coarse grains, respectively, and the coarse-grain reptation threshold $\tau_r(d^{(c)})$ (solid line). The thresholds are calculated from a physical grain-scale model (Methods) for typical terrestrial conditions (kinematic viscosity $\nu_a \approx 1.6 \times 10^{-5}$ m$^2$ s$^{-1}$, atmospheric density $\rho_a \approx 1.2$ kg m$^{-3}$, grain density $\rho_p \approx 2650$ kg m$^{-3}$ and fine-grain size $d^{(f)} \approx 491$ µm, corresponding to Galileo number Ga$^{(f)} \approx 100$ and density ratio $s \approx 2200$). The transport regimes map directly onto dynamical regimes of sand sorting and megaripple evolution, as summarized in Table 1. To generalize this framework to more realistic continuous GSDs, as measured in the field, $d^{(f)}$ is equated to the coarsest saltating grain size, max($d^{(f)}$), at a given wind strength $\tau$. Thereupon, the transport phase diagram collapses onto the thick green line where $\tau_t(\max(d^{(f)})) = \tau$.

**Table 1 Aeolian transport and sorting regimes and megaripple morphodynamics.**

| Transport mode | Wind condition | Coarse grains | Fine grains | Sorting mode | Morphodynamics |
|---|---|---|---|---|---|
| No transport | $\tau < \tau_t(d^{(f)})$ | No motion | No motion | Stagnation | Stagnation |
| Selective transport | $\tau > \tau_t(d^{(f)})$ & $\tau < \tau_r(d^{(c)})$ | No motion | Saltation | Formation | Stagnation |
| Bimodal transport | $\tau > \tau_t(d^{(f)})$ & $\tau_r(d^{(c)}) < \tau < \tau_t(d^{(c)})$ | Reptation | Saltation | Formation | Formation |
| Unimodal transport | $\tau > \tau_t(d^{(c)})$ | Saltation | Saltation | Destruction | Destruction |

An idealized bidisperse sand bed admits four aeolian transport modes that affect sand sorting and megaripple morphology differently (color-code of Fig. 2). *Bimodal transport:* fine grains saltate for shear stress $\tau > \tau_t(d^{(f)})$ and coarse grains reptate for $\tau_r(d^{(c)}) < \tau < \tau_t(d^{(c)})$, causing megaripple migration and growth due to the accumulation of coarse grains. *Unimodal transport:* accordingly, for winds beyond the coarse-grain saltation threshold, the coarse grains of the armouring layer are entrained into saltation, and their thus dramatically increased hop lengths render megaripples subcritical and therefore unstable to erosion. (However, small ripples with a perceptible armouring layer may emerge since even unimodal saltation can lead to spatial sorting[31,36,39].) *Selective transport:* fine-grain impacts cannot mobilize coarse grains ($\tau < \tau_r(d^{(c)})$) and megaripple formation stagnates. Armouring by (immobile) coarse grains can persist due to winnowing (net erosion) of fine grains, which still saltate for $\tau > \tau_t(d^{(f)})$. *No transport:* below the fine-grain saltation threshold $\tau_t(d^{(f)})$, grain transport ceases entirely.

in ref. [38], and partly also in refs. [23–25,40], our phase diagram in Fig. 2 is computed on the basis of a conceptually new, but phenomenologically already well tested, quantitative grain-scale model for the saltation threshold[45]. The latter model is herein generalized to account for the size disparity between saltating fine grains and coarse armouring grains. Specifically, it predicts the saltation threshold values $\Theta_t(d^{(f)})$ and $\Theta_t(d^{(c)})$ (Methods) of the non-dimensionalized wind shear stress (Shields number[46]) $\Theta \equiv \tau/(\rho_p \tilde{g} d)$. The predictions hold for arbitrary aeolian transport conditions, which enter via three scaling parameters: the grain-size ratio $d^{(c)}/d^{(f)}$, the grain-atmosphere density ratio $s \equiv \rho_p/\rho_a$ and the Galileo number $Ga^{(f)} \equiv \sqrt{s\tilde{g}d^{(f)}}d^{(f)}/\nu_a$ (the grain-scale Reynolds number associated with the sedimentation velocity). Here $\tilde{g} \equiv (1 - 1/s)g$ is the buoyancy-reduced gravitational acceleration, $\rho_p$ ($\rho_a$) the grain (atmospheric) density and $\nu_a$ the kinematic viscosity of the atmosphere. For our purpose, the intrinsic grain-scale stochasticity can be averaged out for a typical saltation trajectory, such that an idealized periodic saltation model suffices to provide a physically meaningful, accurate result. Specifically, the model assumes that the saltating grains are driven by buoyancy-reduced gravity and fluid drag caused by an undisturbed mean inner turbulent boundary layer flow (including a potential viscous sublayer). This should be appropriate for the considered predominantly undersaturated conditions associated with megaripple evolution[38]. Grain-bed interactions are simplistically but accurately resolved through mean rebound laws derived in the Supplementary Information for an idealized fixed bed of coarse grains representing the armouring layer on a megaripple (Fig. 1b). The saltation thresholds $\Theta_t(d^{(f)})$ and $\Theta_t(d^{(c)})$ for fine and coarse grains are then defined as the smallest Shields numbers for which a periodic saltation trajectory exists. The reptation threshold $\Theta_r(d^{(c)})$ follows from the momentum balance, assuming a binary fine-grain/coarse-grain collision[38], that is, the collision in which the optimum amount of energy is transferred to the coarse bed grain to make it only just leapfrog over a neighboring bed grain (Supplementary Information). Such rare but optimal impacts require fine grains to have reached a regime of quasi-steady saltation after a long sequence of jumps, which vindicates the simplifying assumptions made in computing $\Theta_r(d^{(c)})$.

**From bidisperse to continuum bimodal sands**. Natural sand is always (to some degree) polydisperse[4] and characterized by a continuous GSD. In this case, the GSD and the size ratio of relevant colliding grains are not fixed a priori but evolve dynamically with the wind strength. This complexity of natural systems contrasts with the underlying simplifying assumption of an idealized bidispersed GSD (Fig. 2). Nevertheless, the transport phase diagram in Fig. 2 can still be applied to natural, polydisperse GSDs. The key insight is that only the impacts of the coarsest saltating grains can supply the large energy required to dislodge the coarsest reptating grains. Hence, the coarsest fine grains that can saltate at a given wind strength dictate, via the coarsest grains they can barely dislodge, the boundaries of the bimodal transport regime—or the size range of (exclusively) reptating grains. We therefore identify the coarsest saltating grains (equivalent to the finest reptating grains) with the fine grains of the idealized bidisperse model, i.e., $\max(d^{(f)}) = d^{(f)}$. This choice of $d^{(f)}$ implies that $\tau/\tau_t(\max(d^{(f)})) = 1$, corresponding to the bold green line in Fig. 2. In this respect, the higher complexity of the natural polydisperse GSD affects the transport process only insofar as the wind itself can now select the coarsest saltating grains — which can thus no longer be externally prescribed. This selection amounts to the "collapse" of the two-dimensional transport phase diagram onto a one-dimensional line in the case of the continuous GSD. Along that line, the bimodal transport regime of the bidisperse model thus delineates

the reptation regime for a continuous GSD, namely the range of its grain sizes that reptate but do not saltate. This range is laterally delimited by the extreme cases of saltation transport at $d^{(c)} = \max(d^{(f)})$ and breakdown of reptation for particles bigger than $\max(d^{(c)})$. The latter is defined by the condition $\tau_r(\max(d^{(c)})) = \tau$, and from Eq. (M3) we find $\max(d^{(c)}) \approx 2.75 \max(d^{(f)})$ under the typical terrestrial conditions assumed in Fig. 2.

Our results suggest that the collapsed transport phase diagram can be directly identified from the GSDs of armoured megaripples (Figs. 3 and S1). Such GSDs usually feature a prominent coarse-grain peak. It was previously argued that this peak can primarily be understood to emerge under erosive conditions, namely, by a winnowing of fine grains that leaves the less mobile coarse grains behind[38]. However, by this argument alone, one would expect the coarsest grains in the right tail of the surface GSD to be those of the underlying bulk GSD. Here, we point out an important additional mechanism that modifies the surface GSD relative to the bulk GSD: the progressive deposition of reptating coarse grains near the crest during megaripple growth. This increases the surface (but not the bulk) concentration of reptating coarse grains independently of any fine-grain erosion. Moreover, it tends to submerge those grains of the bulk distribution that have become exposed at the surface but are too heavy to be dislodged at the prevailing wind strength. Thereby, the surface concentration of reptating coarse grains increases, while that of immobile grains decreases. This leads to the conclusion that the coarse-grain peak of the surface GSD consists of all (recently) reptating grains, so that its abscissa does indeed coincide with the reptation regime of the collapsed phase diagram (Fig. 3). In summary, the left and right margin of the coarse-grain peak in crest GSDs are robustly defined by the maximum grain sizes that can saltate and reptate, respectively. This improves the previous conceptual picture in which only the left margin of the coarse-grain peak was deemed important[38]. Furthermore, it explains why it

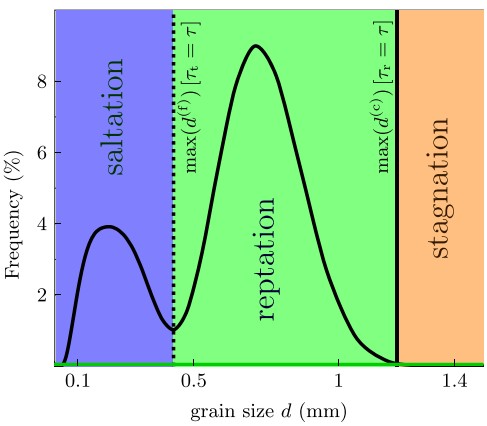

**Fig. 3 Bimodal surface GSD and projected transport modes.** The projection of the phase diagram for bidisperse grain transport from Fig. 2 is intimately linked to the (continuously) bimodal surface GSD found on megaripples (see Fig. S1 for another visual representation of the connection). While the left peak comprises the saltating fine grains and maps to the saltation regime (blue), the right peak comprises the reptating coarse grains and maps to the reptation regime (green). The transition between them corresponds to the minimum of the GSD, where one finds the coarsest saltating fine grains, whose saltation threshold $\tau_t(\max(d^{(f)})) = \tau$ is equal to the prevailing wind stress and constantly adjusts, accordingly (dotted line). Likewise, the transition to stagnation (solid line) defines the coarsest grain size at the right margin of the coarse-grain peak via its reptation threshold $\tau_r(\max(d^{(c)})) = \tau$. The displayed representative surface GSD was obtained from a megaripple crest located in Nahal Kasuy, Israel (see Fig. 1 in ref. [16]).

is insufficient to look for signatures of bimodal transport in the modes of the fine- and coarse-grain peaks (Fig. 1c).

In contrast to the transport regimes, the morphodynamic regimes of megaripple formation and destruction become more complex for natural GSDs as compared to bidisperse GSDs. However, some general statements can be gleaned from the idealized scheme. On the one hand, if the wind strength exceeds the saltation threshold of the coarsest available grains, all grains will saltate and the megaripple will quickly be destroyed. In its place, small ripples with a perceptible armouring layer may emerge, since even unimodal saltation can lead to spatial sorting[31,36,39]. On the other hand, if the wind falls below the saltation thresholds for all available grains, stagnation ensues. Between these limits, the slow coarsening of the megaripple and its armouring layer will potentially compete with partial destruction and stagnation episodes, depending on the wind history. In particular, a slow growth of the armouring layer of very coarse grains can coexist with its rapid shrinking caused by the net erosion of somewhat finer, but previously reptating grains that are suddenly entrained into saltation by a gust. This implies that the GSD's coarse-grain peak permanently adjusts to the prevailing wind strength — with quite diverse rates[38] for coarsening and growth versus erosion and decline, respectively. Their interplay explains the complex and highly variable evolution of the surface GSD due to wind variations, observed in field measurements[22] (Fig. 1a, c).

To summarize, aeolian grain sorting and ensuing structure formation are very sensitive to wind-strength variations. Over geological time scales, the long-term history of wind and sand supply conditions may thereby be recorded in a potentially complex grain-size stratification in the depth-dependent bulk GSD[16]. Compared to small ripples and large dunes[3,27–30], the mid-size megaripples thus stand out as long-lived but transient sand patterns, contingent on the wind history.

**Max-size ratio**. As detailed in the previous section, we found a characteristic signature of the underlying grain-scale dynamics robustly ingrained in the surface GSD of megaripples. Specifically, for a given size $\max(d^{(f)})$ of the coarsest grains of the fine, saltating fraction (left margin of coarse-grain peak), our model predicts the size $\max(d^{(c)})$ of the coarsest grains of the coarse, reptating fraction (right margin of coarse-grain peak) as a function of $s$ and $\mathrm{Ga}^{(f)}$. These dimensionless characteristics of the ambient conditions are typically narrowly defined for any given measurement site. Figure 4 (inset) illustrates that the max-size ratio, $\max(d^{(c)})/\max(d^{(f)})$, increases with increasing $s$ (e.g., decreasing atmospheric density) and $\mathrm{Ga}^{(f)}$ (e.g., increasing fine-grain diameter). This trend arises because the saltating grains are less easily accelerated by winds at high $s$ and $\mathrm{Ga}^{(f)}$, such that increasingly long and more energetic trajectories are needed to sustain saltation.

For sufficiently large fine grains ($s^{1/4}\mathrm{Ga}^{(f)} \gtrsim 200$, see Fig. 6b in ref. [45]), the theoretical prediction simplifies to the following analytical implicit expression (Methods):

$$\frac{\max(d^{(c)})}{\max(d^{(f)})} \approx \left( 0.82\, H \left[\ln\left(\frac{H}{37.74}\right)\right]^{-2} \right)^{1/6}, \tag{1}$$

$$\text{with} \quad H \equiv 40 s V_s^2 \mu_r^2 C_z^2 \frac{\max(d^{(f)})}{\max(d^{(c)})}.$$

Here, $\mu_r C_z = (v_{\downarrow x}^{(f)} - v_{\uparrow x}^{(f)})/(v_{\uparrow x}^{(f)} + v_{\downarrow x}^{(f)})$, with $v_{\uparrow(\downarrow)x}^{(f)}$ the mean horizontal fine-grain rebound (impact) velocity, is a function of $\max(d^{(c)})/\max(d^{(f)})$ (Eq. (M12)), and $V_s$ is the dimensionless settling velocity, which is a known function of $\mathrm{Ga}^{(f)}$ alone (Eq. (M6)). The dimensionless parameter $H$ can be interpreted as the ratio

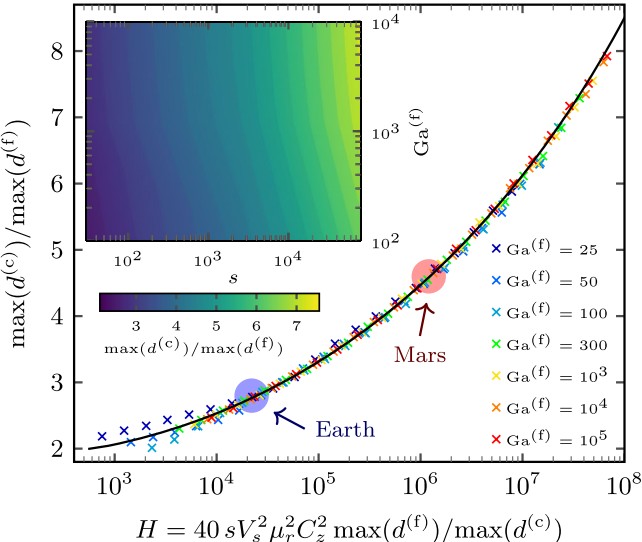

$$H = 40\, s V_s^2 \mu_r^2 C_z^2 \max(d^{(f)})/\max(d^{(c)})$$

**Fig. 4 Solution of the transcendental equations predicting the max-size ratio (general atmospheric conditions).** For given environmental conditions, parametrized in terms of the dimensionless Galileo number $\mathrm{Ga}^{(f)}$ and grain-atmosphere density ratio $s$, reptation is only possible in a specific range of coarse-grain diameters $d^{(c)}$, as encoded in the width of the coarse-grain peak of the surface GSD. For sufficiently large fine grains ($s^{1/4}\mathrm{Ga}^{(f)} \gtrsim 200$), the analytical scaling function in Eq. (1) (solid line), implicitly predicting the max-size ratio $\max(d^{(c)})/\max(d^{(f)})$, provides a perfect match to the full theory (symbols) for various combinations of $\mathrm{Ga}^{(f)}$ and $s$. Note the higher numerical sensitivity to $s$ rather than $\mathrm{Ga}^{(f)}$ (inset) and the breakdown of the scaling for $s^{1/4}\mathrm{Ga}^{(f)} \lesssim 200$. Also, $H$ depends on the planetary conditions primarily via $s$ and the dimensionless settling velocity $V_s$.

between the mean trajectory height and the nominal zero-level $z_0 = d^{(c)}/30$ of the undisturbed log-layer wind profile (under-saturated, rough-bed conditions). The comparison in Fig. 4 reveals how, with decreasing $\mathrm{Ga}^{(f)} \lesssim 100$, the asymptotic analytical equation starts to deviate from our full model since it neglects the viscous sublayer. Note that the latter, which accounts both for the viscous sublayer and log-layer, is itself restricted by the condition for saltation ($s^{1/2}\mathrm{Ga}^{(f)} \gtrsim 200$, $s \gtrsim 100$, see Fig. 6b in ref. [45]).

Interestingly, the power 1/6 in Eq. (1) causes the max-size ratio to respond only very weakly to the environmental conditions encapsulated in $s$ and $\mathrm{Ga}^{(f)}$, varying only between 2.75 on Earth and 4.5 on Mars, below Bagnold's empirical estimate for the modes of the two grain populations[3]. Notably, our model assumes an "optimum" grain-bed collision for the reptation threshold and less efficient collisions would further decrease the max-size ratio. The reason for this insensitivity of the max-size ratio lies in the sensitivity of the energy partition on the mass ratio in binary collisions. In practice, one may prefer to work with the logarithm of the max-size ratio, which is the log-scale width of the coarse-grain peak, $\ln\left(\max(d^{(c)})/\max(d^{(f)})\right) = \ln\left(\max(d^{(c)})\right) - \ln\left(\max(d^{(f)})\right)$. The apparently "unique" max-size ratio $\max(d^{(c)})/\max(d^{(f)})$ and log-scale coarse-grain peak width of the GSD should be contrasted with the sensitivity of the absolute values of $\max(d^{(c)})$ and $\max(d^{(f)})$ to wind-strength variations in order to appreciate the potential benefit of our quantitative analysis for field measurements.

**Comparison to data**. We argued that the surface GSD of megaripple crests is the result of

(i)   the erosion of saltating fine grains,

(ii)   the deposition of reptating coarse grains, and

(iii)  the submersion of the coarsest, immobile grains.

As a crucial consequence, the coarse-grain peak is predicted to have a well-defined width, ingraining the max-size ratio, i.e., the size range of the (recently) reptating grains.

To validate our prediction, we collected data from a wide range of measurement sites (Methods), some corresponding to extreme environmental conditions. For example, in Ladakh, India, the density ratio $s$ is substantially increased due to the high altitude of the plateau (4522 m above sea level), while in Antarctica[15], bedforms composed of coarse gravel experience extreme winds and atmospheric temperatures, and even more extreme conditions prevail on Mars[40]. As detailed in the Methods, the sizes of the coarsest saltating and reptating grains are extracted from the measured GSDs through a local-slope criterion, which allows us to determine the max-size ratio objectively, without the need for parameterization. The model predictions are found to be in good agreement (coefficient of determination $R^2 = 0.91$) with the compiled data (Fig. 5).

According to the above discussion, the max-size ratio should be a stable quantity that is not very sensitive to environmental conditions or modeling inaccuracies, for a given constant wind speed. This is to be contrasted with the precise positions of the two margins of the coarse-grain peak, which are quite sensitive to (recently) prevailing wind strengths, and can therefore even be used to infer the wind circulation from a measured GSD. The response of $\max(d^{(c)})$ to wind variations is more sluggish than that of $\max(d^{(f)})$, which could explain the scatter around the 1:1 line in Fig. 5. In particular, upwards scatter should be indicative

of a recent wind gust causing coarser grains to saltate, thereby narrowing the peak. As noted in ref. [38], the shape of the GSD will saturate only if the wind strength varies slowly compared to the so-called coarsening rate (which is typically quite low). This means that waiting for the incipient formation process to saturate towards a hypothetical stationary regime may be futile, and measurements will always retain a slightly anecdotal character dependent on the actual wind history[24,38,47].

Unfortunately, the lack of standardization in field sampling practices limits our capacity to use field data for model validation (Methods). In particular, measurements generally depend on the precise location and depth of the sand removal from the megaripple[16], which are seldom exactly reproducible. Also, for a reliable measurement of the coarse-grain peak width and the max-size ratio, one needs a sufficiently polydisperse sand source or bulk GSD, comprising some immobile grains at the prevailing wind strength. Otherwise, the right tail of the coarse-grain peak could be cut off by a lack of movable coarse grains. Furthermore, since the log-scale width of the coarse-grain peak will fluctuate somewhat in response to local wind variations, as explained above, more reliable results may be achieved through averaging it over repeated measurements under similar environmental conditions. Finally, the coarsening rate is predicted to be sensitive to the width of the bulk GSD. As a consequence, the saturation of the surface GSD is expected to be substantially faster — and less sensitive to disturbances from brief wind fluctuations — for more evolved megaripples with (typically) thicker armouring layers, which usually reside on more polydisperse bulk sands. These are moreover morphologically more stable[24], making them good candidates for long-term observations.

## Discussion

We found that a characteristic signature of grain-scale transport is encoded in the grain-size distributions (GSDs) that co-evolve with megaripples. Our compilation of original and literature data[6,15,16,20,22,40] firmly establishes the accuracy and robustness of the theoretical prediction across a wide range of geographic locations and prevailing environmental conditions. Laboratory data as well as field data from Antarctica and Opportunity rover data from Mars were found to obey the same parameter-free functional relationship as data measured in major sand deserts around the globe. Our primary finding is that the margins of the coarse-grain peak of the bimodal GSDs on armoured megaripple crests can directly be associated with the coarsest saltating grains of the fine-grain fraction and the coarsest (still mobile) reptating grains of the coarse-grain fraction, respectively. This link between the GSD and the underlying grain-scale transport mechanism is robust against considerable deformations of the GSD and the emergent topography, as caused by variable wind strengths. And, remarkably, the same holds for the so-called max-size ratio of the coarsest mobile grains of each fraction and the log-scale width of the coarse-grain peak of the GSD itself. This observation explains why important insights may already be gained from the discussion of a schematic GSD. Albeit, we have shown the individual margins of the coarse-grain peak of empirical GSDs to be sensitive to the most recent wind strength, which especially affects the size of the coarsest saltating grains. The latter should thus provide additional information about the prevailing wind strength, provided an appropriate supply of coarse immobile grains. The theory's ability to reveal useful information about the environmental conditions from such a peculiar mix of robust and sensitive traits encoded in empirical megaripple GSDs makes it a potentially powerful practical tool to asses wind conditions on other planetary bodies. Likewise, we anticipate that our prediction for the log-scale width (Eq. (1)) could shed light onto the current debate about the origin of various kinds of mid-sized bedforms on Mars[48–50]. Compared to the mere existence of bimodal surface GSDs[40,51–56], it provides a more stringent,

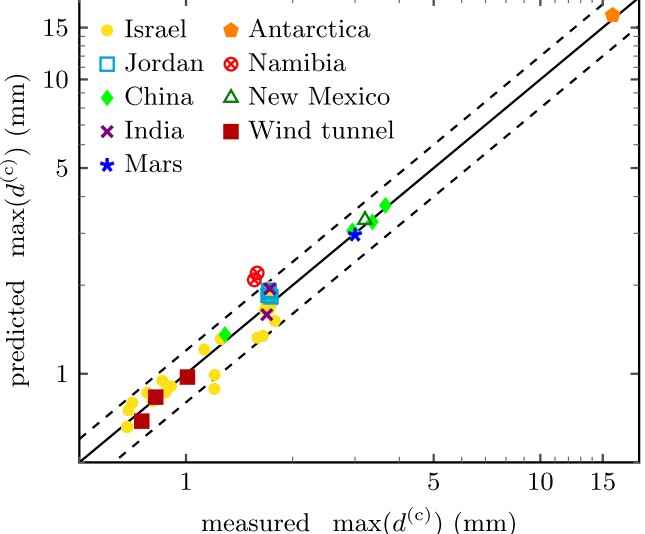

**Fig. 5 Comparison between data and predictions.** The coarsest grain sizes $\max(d^{(f)})$ and $\max(d^{(c)})$ are extracted from the left and right margins, respectively, of the coarse-grain peaks of the empirical GSDs (Methods). For a given measurement site, with a given measured size $\max(d^{(f)})$ of the coarsest saltating grains, the theoretically predicted size $\max(d^{(c)})$ of the coarsest reptating grains is plotted against its measured value. The solid line corresponds to perfect agreement with the idealized model, and the dashed lines indicate a relative error of 20%. See main text for plausible origins of the scatter in the data and suggestions for more accurate measurement protocols. The displayed data is from megaripples located at Nahal Kasuy, Ktora and Yahel in the southern Negev, Israel (filled circles)[16,20,22] (Figs. S4–6); Wadi Rum in southern Jordan (open squares)[16]; Sanshan Desert in western China (filled diamonds)[6,16]; Antarctica (filled pentagons)[15]; Sossusvlei in Namibia (crossed circles) (Fig. S7); Ladakh in India (crosses) (Fig. S8); New Mexico (open triangles)[40]; and Mars (stars)[40] (Methods). Wind tunnel data are from ref. [20].

quantitative criterion to discern between megaripples and reminiscent patterns composed of unimodal sands[48–50,57,58], or other bedforms[59,60]. For example, we have used an image of a Martian GSD from the Opportunity Rover[40] for our model validation in Fig. 5. The fact that the corresponding measurement point falls on the prediction is a strong indication that the image was taken on the surface of a megaripple. On the other hand, if future missions were to find evidence of bimodal GSDs with a larger coarse-grain width than predicted, it would be strongly suggestive for bedforms with different mechanistic origins. For example, spatial grain sorting and bedform evolution occurs for bidisperse sands even when both grain sizes are saltating[31,36,39], so that not all bedforms with a perceptible armouring layer on the crest need to be megaripples in the narrow sense of the term used here. If instead future missions were to find evidence of bimodal GSDs with smaller-than-predicted coarse-grain peaks, megaripples could not be completely ruled out because of the possibility of a total lack of immobile coarse grains.

Slight extensions of our theoretical framework — as well as of field measurements to test it — will likely be needed for sands comprising grains of different mass density, as reported for megaripples in Argentina Puna[17–19,61] and (possibly) Lybia[62]. Other promising directions to pursue in the future concern the validation of our predictions using low-pressure wind tunnels, the introduction of stochastic elements into the underlying deterministic periodic saltation model, and its extension to so-called bedload transport, in which grain-bed impacts can no longer be considered as isolated events[45,63].

## Methods

Here, we provide the methods and technical details to

(i) quantify the transport phase diagram of bidisperse sand by means of reptation and saltation thresholds,
(ii) predict the width of the coarse-grain peak of megaripple GSDs and
(iii) test the theory against empirical data.

Note that the mathematical symbols appearing in this Methods section are summarized in Table S1 and the Notation section of ref. [45].

**Reptation threshold.** Megaripple formation and growth are associated with the bimodal transport regime in the transport phase diagram for bidisperse sands (Fig. 2). One of its ingredients is the wind strength at the onset of coarse-grain reptation. Beyond the crude semi-empirical estimate of this reptation threshold shear stress $\tau_r(d^{(c)})$ given in ref. [38], we here provide a fully quantitative prediction based on a precise modeling of the underlying grain-scale physics (chiefly based on the conservation laws).

The basic idea is that the reptation of coarse grains of diameter $d^{(c)}$ is driven by kicks from saltating fine grains of diameter $d^{(f)}$, which are hopping over an armoring layer of such coarse grains (Fig. 1b). Similar to ref. [38], we estimate $\tau_r(d^{(c)})$ from an extreme (optimum) binary collision between a fine grain of mass $m^{(f)}$ and a coarse grain of mass $m^{(c)}$, in which the maximum possible kinetic energy $E^{(c)}$ is transmitted. Momentum balance in a head-on collision yields[38]

$$E^{(c)} = \frac{1}{2} m^{(f)} \left(\frac{m^{(f)}}{m^{(c)}}\right)^2 (\Delta \mathbf{v}^{(f)})^2,$$ (M1)

where $m^{(f)} \Delta \mathbf{v}^{(f)} = m^{(f)}(\mathbf{v}_\downarrow^{(f)} - \mathbf{v}_\uparrow^{(f)})$ is the momentum change of the fine grain between its impact with velocity $\mathbf{v}_\downarrow^{(f)}$ and its rebound with velocity $\mathbf{v}_\uparrow^{(f)}$. The minimum energy $E_{crit}$ needed to dislodge the coarse grain out of its position in the bed is derived in the Supplementary Information. For a hexagonal arrangement of armouring grains, we find

$$E_{crit} = 0.326 \, m^{(c)} \tilde{g} d^{(c)}.$$ (M2)

Combining Eqs. (M1) and (M2), the non-dimensionalized fine-grain velocity difference $\Delta \tilde{\mathbf{v}}^{(f)} = \Delta \mathbf{v}^{(f)} / \sqrt{s \tilde{g} d^{(f)}}$ for such critical conditions is

$$\left(\Delta \tilde{\mathbf{v}}^{(f)}\right)^2 = 0.652 \, s^{-1} \left(\frac{d^{(c)}}{d^{(f)}}\right)^7.$$ (M3)

The reptation threshold $\Theta_r(d^{(c)})$ can now be calculated as the smallest Shields number $\Theta$ at which fine-grain saltation reaches this critical velocity difference. As mentioned in the main text, $\Theta_r(d^{(c)})$ depends on environmental conditions in dimensionless form via the grain-atmosphere density ratio $s$, the grain-size ratio $d^{(c)}/d^{(f)}$, and the Galileo number $Ga^{(f)}$. For its computation, the fine-grain motion is idealized by a periodic saltation trajectory over a coarse-grain bed. While the original periodic saltation model[45] was constructed for monodisperse sand, we here account for the size disparity between the saltating fine grains and the coarse bed

grains (see below and subsequently Supplementary Information for a generalized rebound law).

**Saltation thresholds.** We here outline the computation of the saltation thresholds of fine and coarse grains needed to construct the phase diagram of transport modes for bidisperse sands in Fig. 2. In recent years, progress has been made in understanding the physics behind the cessation (or "impact") threshold of saltation, i.e., the wind stress below which an ongoing saltation process dies out[41]. While earlier studies have associated it with the splash ejection of bed grains[42,64–70], recent studies favor the more accurate notion of a "rebound threshold"[41,45,63,71–73]: the minimum wind strength needed to compensate via wind drag acceleration the average energy dissipated during grain-bed rebounds. Based on this interpretation, Pähtz et al.[45] proposed a periodic saltation model that is in agreement with a large body of measurements and grain-scale simulations across aeolian and fluvial sediment transport conditions. The model idealizes the grain motion by periodic saltation trajectories (see next subsection). Given a vertical lift-off velocity $\hat{v}_{\uparrow z}^{(f,c)}$ (in units of the settling velocity of fine or coarse grains, respectively), we calculate the unique corresponding Shields number $\Theta^{(f,c)}$ that allows for periodic saltation. Its smallest value—i.e., the smallest possible wind shear stress for which a periodic saltation trajectory can be sustained—is then identified with the rebound saltation threshold:

$$\Theta_t^{(f,c)} \equiv \min_{\hat{v}_{\uparrow z}^{(f,c)}} \Theta^{(f,c)} \left(Ga^{(f,c)}, s, d^{(f,c)}/d^{(f,c)}, \hat{v}_{\uparrow z}^{(f,c)}\right).$$ (M4)

While we can reasonably assume monodisperse sand to compute the saltation threshold for the coarse grains, it is necessary to account for the more efficient rebound of fine grains hopping on the armouring layer, which leads to a different value of the fine-grain saltation threshold compared to monodisperse saltation threshold[45] (Fig. 6). Interestingly, we find that its grain-size dependence is non-monotonic: with increasing $d^{(c)}/d^{(f)}$, $\Theta_t^{(f)}$ first decreases, thanks to the more energetic rebounds, and then increases, due to increasing $z_0/d^{(f)}$.

**Periodic saltation model.** Here, we briefly summarize relevant aspects of the periodic saltation model[45] used to quantify the motion of the saltating grains in the above threshold calculations and to analytically estimate the width of the reptation regime. While its original version was derived for monodisperse sands ($d^{(c)} = d^{(f)}$), we here extend it to also cover the scenario of fine-grain saltation over an armouring layer of coarse grains ($d^{(c)} \geq d^{(f)}$). The stochastic nature of the transport process (due to turbulence, surface inhomogeneities, and bed arrangement[69]) is neglected, since the motion of grains with size $d^{(f)}$ is (effectively) represented by a deterministic periodic saltation motion driven by a mean wind velocity, undisturbed by the grain. However, there are good physical arguments for this modeling strategy, and it has successfully been employed in similar contexts[45,65,71,74,75]. In particular, neglecting grain-wind feedback is justified near the threshold[45] and more generally for sufficiently undersaturated transport conditions, as expected in the context of megaripple formation[38].

The wind velocity profile $u_x(z)$ is approximated by a mean inner turbulent boundary layer flow above a flat wall mimicking the sand bed composed of coarse

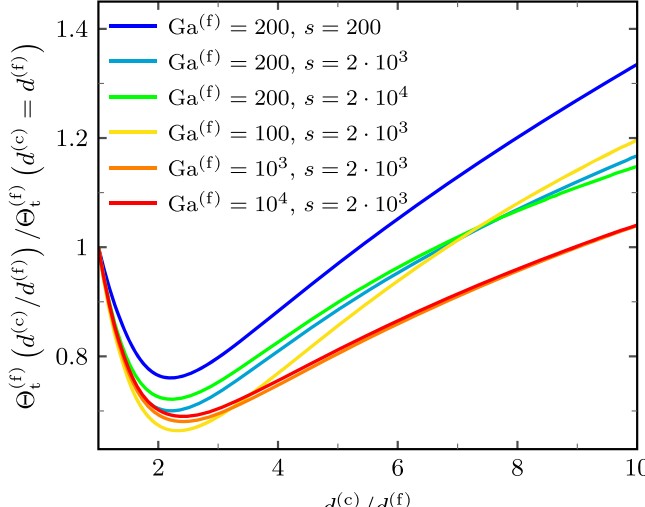

**Fig. 6 Relation between bidisperse and monodisperse saltation thresholds.** Comparison of fine-grain saltation thresholds based on the periodic saltation model for bidisperse, $\Theta_t^{(f)}(d^{(c)}/d^{(f)})$, and monodisperse, $\Theta_t^{(f)}(d^{(c)} = d^{(f)})$, sand for different atmospheric conditions parametrized by the Galileo number $Ga^{(f)}$ and density ratio $s$.

grains. It consists of three potential sublayers: a linear, viscous sublayer just above the surface (except for rough surfaces), a logarithmic layer sufficiently far away from the surface, and a buffer layer connecting the two. The entire profile is parametrized in terms of the coarse-grain shear Reynolds number ($\mathrm{Re}_*^{(c)} = \mathrm{Ga}^{(f)}\sqrt{\Theta^{(f)}}d^{(c)}/d^{(f)}$) as[76,77]

$$
\begin{aligned}
u_x\sqrt{\frac{\rho_a}{\tau}} &= f_u(\mathrm{Re}_*^{(c)}, z/d^{(c)}) \\
&\equiv 7\ \arctan\left[\mathrm{Re}_*^{(c)}(z/d^{(c)} + Z_\Delta)/7\right] \\
&+ \frac{7}{3}\arctan^3\left[\mathrm{Re}_*^{(c)}(z/d^{(c)} + Z_\Delta)/7\right] \\
&- 0.52\arctan^4\left[\mathrm{Re}_*^{(c)}(z/d^{(c)} + Z_\Delta)/7\right] \\
&+ \ln\left\{1 + \left[\mathrm{Re}_*^{(c)}(z/d^{(c)} + Z_\Delta)/B_\kappa\right]^{1/\kappa}\right\} \\
&- \frac{1}{\kappa}\ln\left[1 + 0.3\,\mathrm{Re}_*^{(c)}\left(1 - e^{-\mathrm{Re}_*^{(c)}/26}\right)\right],
\end{aligned}
\tag{M5}
$$

where $Z_\Delta = 0.2 + 0.5d^{(f)}/d^{(c)}$ is the average dimensionless elevation of grain-bed rebounds and $B_\kappa \equiv \exp(16.873\kappa - \ln 9)$, with $\kappa = 0.4$ the von Kármán constant.

For the fluid-grain interactions, we consider only the fluid drag and buoyancy force, yielding deterministic laws directly mapping the rebound velocity $\mathbf{v}_\uparrow^{(f)}$ to the impact velocity $\mathbf{v}_\downarrow^{(f)}$. The effect of the fluid drag is encoded in the (dimensionless) settling velocity:

$$
\begin{aligned}
V_s &\equiv \frac{v_s}{\sqrt{s\tilde{g}d^{(f)}}} = \frac{\overline{u}_x - \overline{v}_x^{(f)}}{\mu_r\sqrt{s\tilde{g}d^{(f)}}} \\
&= \frac{1}{\mu_r}\left[\sqrt{\sqrt[m]{\frac{1}{4}\left(\frac{24}{C_d^\infty \mathrm{Ga}^{(f)}}\right)^2} + \sqrt[m]{\frac{4\mu_r}{3C_d^\infty}}}\right. \\
&\left. - \frac{1}{2}\sqrt[m]{\frac{24}{C_d^\infty \mathrm{Ga}^{(f)}}}\right]^m,
\end{aligned}
\tag{M6}
$$

where $C_d^\infty = 0.4$ and $m = 2$ (for spherical grains), and $\mu_r$ is a rebound momentum restitution coefficient:

$$
\mu_r \equiv \frac{v_{\downarrow x}^{(f)} - v_{\uparrow x}^{(f)}}{v_{\uparrow z}^{(f)} - v_{\downarrow z}^{(f)}}.
\tag{M7}
$$

The solution for the vertical motion is (in dimensionless form)

$$
\hat{v}_{\downarrow z}^{(f)} = \frac{v_{\downarrow z}^{(f)}}{\sqrt{V_s s\tilde{g}d^{(f)}}} = -1 - W\left[-(1 + \hat{v}_{\uparrow z}^{(f)})e^{-(1 + \hat{v}_{\uparrow z}^{(f)})}\right],
\tag{M8}
$$

where $W$ denotes the principal branch of the Lambert-$W$ function. Finally, the Shields number corresponding to the periodic trajectory solution can be expressed as

$$
\Theta^{(f)} = \frac{\sqrt{\Theta^{(f)}}V_s[\mu_r(1 + \hat{v}_{\uparrow z}^{(f)}) + \hat{v}_{\uparrow x}^{(f)}]}{f_u\left\{\mathrm{Re}_*^{(c)}, sV_s^2\frac{d^{(f)}}{d^{(c)}}\left[-\hat{v}_{\downarrow z}^{(f)}(1 + \hat{v}_{\uparrow z}^{(f)}) - \hat{v}_{\uparrow z}^{(f)}\right]\right\}}.
\tag{M9}
$$

To close the model, we have to prescribe the grain-bed rebounds to connect the impact velocity $\mathbf{v}_\downarrow^{(f)}$ back to the rebound velocity $\mathbf{v}_\uparrow^{(f)}$. Going beyond the phenomenological characterization for the special case of $d^{(c)} = d^{(f)}$ in ref.[45], we characterize the rebound for arbitrary grain-size ratios within an improved analytical version of the detailed bottom-up rebound model of ref.[78] (Section S2). It allows the impact and rebound fine-grain velocities to be computed from the average total and vertical restitution coefficients

$$
\overline{e} \equiv |\mathbf{v}_\uparrow^{(f)}|/|\mathbf{v}_\downarrow^{(f)}|,
\tag{M10a}
$$

$$
\overline{e_z} \equiv -v_{\uparrow z}^{(f)}/v_{\downarrow z}^{(f)},
\tag{M10b}
$$

which both are functions of the average impact angle and grain-size ratio $d^{(c)}/d^{(f)}$.

In summary, the solution of Eqs. (M5)–(M10) determines the family of periodic trajectories, parametrized only by the (dimensionless) wind strength $\Theta^{(f)}[\mathrm{Ga}^{(f)}, s, d^{(c)}/d^{(f)}, \hat{v}_{\uparrow z}^{(f)}]$.

**Analytical max-size ratio.** To obtain the max-size ratio $\max(d^{(c)})/\max(d^{(f)})$ and its scaling with ambient parameters in the closed form presented in Eq. (1), the general reptation threshold model described above is further simplified. Restricting the periodic saltation model to the limiting regime of "turbulent saltation" (i.e. $s^{1/4}\mathrm{Ga}^{(f)} \gtrsim 200$, see Fig. 6b in ref.[45]), an analytical prediction for the velocity difference $\Delta\mathbf{v}^{(f)}$ and the fine-grain saltation threshold can be worked out as follows. First, for turbulent saltation, the flow velocity profile can be approximated by that for the log-scale boundary layer with $Z_\Delta \to 0$, so that Eq. (M5) becomes

$$
f_u(\mathrm{Re}_*^{(c)}, z) \sim f_u(\mathrm{Re}_*^{(c)} \to \infty, z) \simeq \frac{1}{\kappa}\ln\left(\frac{z}{z_0}\right),
\tag{M11}
$$

where $z_0 = d^{(c)}/30$ characterizes the nominal zero-velocity elevation for an undisturbed rough aerodynamic boundary layer. Secondly, at the turbulent saltation threshold, taking the limit of negligible vertical drag has almost no effect on the final prediction (see Fig. 6 in ref.[45]), so that the kinetic energy of the grain's vertical motion is conserved (corresponding to $\overline{e_z} \equiv -v_{\uparrow z}^{(f)}/v_{\downarrow z}^{(f)} = 1$). One then has a fixed impact angle, a total restitution coefficient $\overline{e} = \overline{e}(\theta_\downarrow^{(f)}, d^{(c)}/d^{(f)})$ and a mean horizontal velocity[45] $\overline{v}_x^{(f)} = (v_{\uparrow x}^{(f)} + v_{\downarrow x}^{(f)})/2$. As a consequence, all required observables can be expressed as functions of $\overline{v}_x^{(f)}$ and the grain-size ratio:

$$
v_{\uparrow z}^{(f)} \equiv C_z\overline{v}_x^{(f)},
\tag{M12a}
$$

$$
|\Delta\mathbf{v}^{(f)}| \equiv C_\Delta\overline{v}_x^{(f)},
\tag{M12b}
$$

$$
\mu_r = \frac{\sqrt{C_\Delta^2 - (2C_z)^2}}{2C_z},
\tag{M12c}
$$

$$
\mu_r C_z = \frac{v_{\downarrow x}^{(f)} - v_{\uparrow x}^{(f)}}{v_{\uparrow x}^{(f)} + v_{\downarrow x}^{(f)}} \le 1,
\tag{M12d}
$$

with the proportionality constants well approximated by

$$
C_z \approx 1.47 - \frac{4.34}{3.45 + 0.47\left(\frac{d^{(c)}}{d^{(f)}}\right)^3},
\tag{M13a}
$$

$$
C_\Delta \approx 3.56 - \frac{4.33}{1.48 + 0.16\left(\frac{d^{(c)}}{d^{(f)}}\right)^3}.
\tag{M13b}
$$

Furthermore, in the limit of negligible vertical drag, the mean value of the wind velocity of the turbulent boundary layer can be estimated by the wind velocity at the mean trajectory height (see Appendix F in ref.[45]):

$$
\overline{u}_x \approx u_x(\overline{z}) = \frac{\sqrt{s\tilde{g}d^{(f)}}\sqrt{\Theta^{(f)}}}{\kappa}\ln\left(\frac{\overline{z}}{z_0}\right),
\tag{M14}
$$

with

$$
\overline{z} = V_s^2 sd^{(f)}\frac{\hat{v}_{\uparrow z}^2}{3}.
\tag{M15}
$$

Using Eqs. (M12)–(M15) we can interpret Eq. (M6) as an implicit expression for $\Theta^{(f)}$ as a function of $\overline{v}_x^{(f)}$. Minimizing $\Theta^{(f)}$ in this expression yields (see also Eq. S26 in the Supplemental Material of ref.[79]),

$$
\overline{v}_x^{(f)} = \sqrt{s\tilde{g}d^{(f)}}\frac{2}{\kappa}\sqrt{\Theta_t^{(f)}}.
\tag{M16}
$$

Setting $\tau_r(\max(d^{(c)})) = \tau = \tau_t(\max(d^{(f)}))$, Eq. (M3) can be rearranged to

$$
\left(\frac{\max(d^{(c)})}{\max(d^{(f)})}\right)^7 \approx 38.34sC_\Delta^2\Theta_t^{(f)}.
\tag{M17}
$$

One can explicitly solve for the fine-grain saltation threshold by inserting Eqs. (M12)–(M16) in Eq. (M6) (see analogously Section 3.2 in ref.[45]), obtaining:

$$
\Theta_t^{(f)} \approx B^{-1}\exp\left[2W\left(\frac{\sqrt{AB}}{2e}\right) + 2\right],
\tag{M18}
$$

with

$$
A = \kappa^2 V_s^2 \mu_r^2, \qquad B = 30s\frac{4C_z^2}{3\kappa^2}\frac{d^{(f)}}{d^{(c)}},
\tag{M19}
$$

again using the Lambert-W function, which takes the asymptotic form $W(x) \sim \ln(x) - \ln(\ln x)$ for large arguments. Hence,

$$
\Theta_t^{(f)} \sim A\left[\ln\left(\frac{AB}{4e^2}\right)\right]^{-2},
\tag{M20}
$$

which finally yields Eq. (1) of the main text.

**Empirical data.** As detailed in the Results, the width of the reptation regime in the (collapsed) phase diagram directly maps to the width of the coarse-grain peak of the bimodal GSD. To compare the theoretical prediction in Eq. (M17) with field observations, we compiled a data set from our own original field measurements and a survey of the literature.

Original field data for the crest GSDs are from the southern Negev (Nahal Kasuy, Yahel and Ktora) in Israel (Figs. S4–6), Sossusvlei in Namibia (Fig. S7) and Ladakh in India (Fig. S8). Each GSD was obtained from a sample taken locally from the crest of a megaripple. Samples of grains were retrieved using the technique described in ref.[16]. We first drained the sand with water to stabilize the megaripple, cut it, and collected the samples about 30 mm below the megaripple apex by using a tin can. All GSDs were obtained from the samples using a high-resolution laser

diffractometer technique (ANALYSETTE 22 MicroTec Plus). It covers the grain-size range 0.04 μm to 1924 μm with a resolution of 102 bins of increasing width $\Delta d$ from $\Delta d = 0.05$ μm in the very fine fraction to $\Delta d = 183$ μm in the very coarse fraction. The raw data can be found in Supplementary Data 1.

Further, we scoured the literature for suitable GSDs[3–6,8,11–13,15–18,20–22,25,35,40,53,62,80–85]. Studies reporting either only two (or even just one) characteristic grain sizes[11,12,17,18,62,80,81,83], or with GSDs that did not contain the full coarse-grain peak[8,85], or with insufficient resolution (grain sizes discretized in class intervals larger or equal than $-0.5 \cdot \log_2 d$)[3,4,13,82], or with too low (single-digit) counts of grains in the coarse-grain peak[53], especially in the right tail of the coarse-grain peak, were sorted out. We also could not use data from ref. [21], since each sample contained material from several bedforms, and data from ref. [84], for which we were unsure whether the location of the right margin of the coarse-grain peak originates from the reptation regime or is due to the absence of bigger immobile grains in the sand source. Data from the remaining studies were pooled according to geographic locations: A large fraction of the field data was collected by the authors in southern Negev, Israel and southern Jordan. We used the GSDs from the following studies (Nahal Kasuy: Fig. 1 (green line) and Fig. 9 in ref. [16], Fig. 6 in ref. [22], Fig. 3 in ref. [20]; Ktora: Fig. 1 (red line) in ref. [16]; Wadi Rum: Fig. 1 (yellow line) and Fig. 10 in ref. [16]. Also GSDs from Shanshan desert, China could be used: one extracted from Fig. 1 (blue line) of ref. [16] and four extracted from Fig. 4 in ref. [6] (red points). From ref. [40], we used two GSDs: one taken from the crest of a megaripple found at the White Sands National Monument, New Mexico (their Fig. 10, solid line), and one acquired by performing a grain-size analysis of a "Microscopic Imager" image of the surfaces of a megaripple located in the Meridiani Planum on Mars (their Fig. 7a). Unfortunately, the resolution of the GSDs (especially the right part of the coarse-grain peak) of sediment samples from the Wright Valley bedforms in Antarctica[15] was not high enough for our analysis (see next subsection). But the measurement data for the grain movement could be used. As the saltation trap data signify that the largest grain size in saltation was between 4 mm and 5.65 mm, we interpreted the mean value as size of the coarsest saltating grain ($\max(d^{(f)}) \approx 4.83$ mm). Similarly we estimated the largest grain size in reptation as the size of the largest grains found in the traction trap ($\max(d^{(c)}) \approx 16$ mm). Lastly, we use the GSDs acquired from experiments conducted in the stationary boundary layer wind tunnel of the Aeolian Simulation Laboratory at Ben-Gurion University, Israel (Fig. 4 in ref. [20]).

For each location on Earth, the air density and kinematic viscosity were calculated with the help of the "1976 Standard Atmosphere Calculator" (https://www.digitaldutch.com/atmoscalc/). The Martian atmospheric kinematic viscosity and density were taken from Table 1 in ref. [40]. Numerical values are listed in Table S2.

**Data extraction**. For the comparison with empirical data, the left and right margin of the coarse-grain peak — corresponding to the coarsest saltating $\max(d^{(f)})$ and the coarsest reptating grains $\max(d^{(c)})$, respectively — has to be extracted from the measured bimodal GSD in a reproducible way. One might try to fit the coarse-grain peak with an appropriate function. In view of the slow grain-sorting time scale[38], only the more or less Gaussian long-time statistics (as opposed to their small-scale Weibull-like statistics[86–89]) of the turbulent wind fluctuations matters, suggesting to fit the peak with a normal distribution of mean $\mu$ and standard deviation $\sigma$. Unfortunately, as the form of the coarse-grain peak is a complex function of the saltation flux of fine grains, the reptation flux of coarse grains, the bulk sand composition, and especially the wind fluctuations over a range of frequencies, its shape is volatile and variable. This speaks in favor of a more local criterion for extracting the max-size ratio from the coarse-grain peak, more focused on its tails, which, after all, are the very traits of the peak tied to the physical transport mechanism through the corresponding aeolian thresholds. Furthermore, it is an intrinsic property of the grain sorting process that it acts only on the relative grain size[38], i.e., on the slope of the GSD. For these reasons, we chose to determine the values $\max(d^{(f)})$ and $\max(d^{(c)})$ as those for which the derivative $f'$ of the GSD corresponds to that of a normal distribution evaluated at $\mu \pm 3\sigma$:

$$|f'(\max(d^{(f)}))| \equiv |f'(\max(d^{(c)}))|$$

$$= \frac{2 \cdot 3^2 e^{-\frac{3^2}{2}}}{\sqrt{2\pi}\left[\max(d^{(c)}) - \max(d^{(f)})\right]^2} . \qquad (M21)$$

Since this extraction method is based on a local criterion, the measured GSDs have to be interpolated and filtered first to get robust results. The numerical values can be found in the Table S2.

## Data availability

All data generated in this study are either extracted from other studies[3,6,13,15,16,20–22,40,82,84,85] or generated from our own measurements. They are summarized in the Supplementary Information (Tables S2, S3 and Figs. S4–8). Source data are provided with this paper.

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

## Acknowledgements

This research was supported by a grant from the GIF, the German-Israeli Foundation for Scientific Research and Development (no. 155-301.10/2018). T.P. acknowledges support from grant Young Scientific Innovation Research Project of Zhejiang University (no. 529001*17221012108).

## Author contributions

K.T. and K.K. devised the study; K.T. and T.P. performed the underlying grain-scale modeling; H.Y. and I.K. designed and conducted the experiments and field surveys; K.T. and I.K. analyzed the data; K.T., T.P., and K.K. wrote the paper. All authors discussed the results and implications and commented on the manuscript at all stages.

## Funding

## Competing interests

The authors declare no competing interests.
