## [Peer Review File · Nature Communications]

Reviewers' Comments:

Reviewer #1:

Remarks to the Author:

My opinion about this paper: Great science, thanks for doing this great work. However, I believe that the next time you should try writing a little more to the point, to facilitate the readers' life.

1 Following this overall assessment, my major criticism is that you should acknowledge others' successful megaripple models. If I remember right, the first successful simulation of megaripples was published by Anderson and Bunas (1993), which is Ref. 31 in Lämmel et al. (2018) (i.e., Ref. 31 of your Ref. 31). As you know, they implemented the idea of megaripple formation and inverse grading due to differential hop lengths of fine/coarse reptating particles, and their model inspired other successful models as well, such as Makse (2000), not cited in your paper, but cited in Lämmel et al. (2018) as Ref. 32. Moreover, I think Yizhaq (2004) is also inspired by Anderson and Bunas (1993) to some extent, but with the difference that in Yizhaq (2004) (as well as in Lämmel et al. (2018) and in your paper) the fine grains saltate and the coarse grains reptate.

The reader should be informed what your work is adding to the model of Anderson and Bunas (1993), and your paper should acknowledge this preceding work.

2 Building on my comment above, the curious reader might raise the question whether it is possible to reconcile Fig. 5 of your paper with the great model of Anderson and Bunas (1993). After all, you don't need a (strict) bimodal sand model for Fig. 5 to be valid, right? It would be a great add to the manuscript if you explicitly address this question with a discussion. I think you already do so at different parts of the manuscript, but doing so more explicitly by referring to Anderson and Bunas (1993) would be pertinent owing to the historical relevance of their work.

3 Line 174 - 179: Do you mean that the only new result is Fig. 5 then?

4 Fig. 3: The labels on the y-axis are missing, "Frequency (%)" - where are the values of the frequency?

5 Line 470 - 474: I fully agree. However, I can wholeheartedly recommend the authors to add - if possible - somewhere in the paper an alternative version of Eq. (1) where the most critical parameters, V_s , μ_r , C_z , appear as a function of the nondimensional numbers (Galileo, Reynolds, Stokes), and physical parameters/constants (gravity, dielectric constant, whatsoever), so that the field/planetary geomorphologist will be able to more easily put all numbers in the calculator and use your Nat Comm paper in remote sensing missions. This is a recommendation that is not critical for acceptance, so if you see that there is no way to do a good job here (which would be unfortunate), then you can disregard this comment.

A bonus comment to round up the discussion: Is this work somehow contributing relevant knowledge to understand the origin(s) of the transverse megaripple instability?

Anderson, R. S. & Bunas, K. L. 1993. Grain size segregation and stratigraphy in aeolian ripples modelled with a cellular automaton. *Nature*, 365, 740-743.

Makse, H. A. 2000. Grain segregation mechanism in aeolian sand ripples. *The European Physical Journal E*, 1, 127-135.

Yizhaq, H. 2004. A simple model of aeolian megaripples. *Physica A: Statistical Mechanics and its Applications*, 338, 211-217.

Reviewer #2:

Remarks to the Author:

Summary. Megaripples are eolian bedforms found in windblown deserts of Earth and Mars. They are characterized by polydisperse grain-size distributions, with coarser grains concentrated near the bedform crests, and that typically display longer wavelengths than the smaller impact ripples. In their manuscript, Tholen et al. develop a model to explain the grain size distribution (GSD) at the crests of mature megaripples through bimodal transport, with finer grains moving in saltation and coarser grains moving in reptation. A novel result from the model is that the width of the coarse-grain mode in the GSD of megaripple crests encodes information about formative winds in a predictive fashion, which according to the authors may have applications in remote-sensing studies of both terrestrial and extraterrestrial surfaces.

Overall Evaluation. The manuscript reports on an interesting model prediction which could possibly have applications as a (paleo)environmental indicator. The model formulation is skillful, and figures are of overall high quality. However, I noted several important weaknesses in the manuscript that, in my opinion, would preclude the manuscript from publication in Nature Communications in its current form:

(1) Novelty: It is not immediately clear how the paper's framework is different from the conceptual model of Lammell et al. (2018). Figure 2 directly maps onto the latter model: "a window of wind strengths can be identified, in which the coarse grains do not themselves saltate but merely creep or 'reptate' over the sand bed, slowly driven by the saltating fraction of fine grains. This wind range is delimited by the threshold wind shear stresses τ_{ct} and τ_{cr} for saltation and reptation of the coarse grains and thereby sensitive to the ratio a_c/a_f of the characteristic coarse and fine sand grain diameters a_c and a_f " (direct quote from Lammell et al., 2018). As it stands, the conceptual model appears to be merely the same as that of Lammell et al., using a different parameterization of transport modes, and with one added prediction (that of the max width). To claim novelty and warrant publication in Nature Communications, I would strongly encourage the authors to clarify and emphasize the distinction between Lammell et al. (2018) and this manuscript, and the specific contributions of this study.

(2) Model Description: The methods section is very helpful; however, the manuscript as it stands does not provide the reader with any information on what physics went into producing the results in the main text. I would strongly recommend that the authors include a bit more information about the "ingredients" that went into the model in the main text. This does not need to be a detailed discussion as the Methods does a great job at that – just enough information for the reader to have some idea of what went into the model without having to dig into the full mathematical formulation.

(3) Parameterization of Wind Profile: The wind profile is parameterized using a z_0 formulation that is appropriate for aerodynamically rough beds with no active transport. However, unless I am missing something, the modeled wind profile is intended to be representative of wind events that transport the fine fraction in saltation, and should thus be a function of the height of the saltation layer rather than grain roughness. The authors should at the very least justify their use of an aerodynamically rough/no transport formulation of z_0 . Note also that martian winds, when no active transport occurs, are not so clearly aerodynamically rough (roughness-scale Reynolds number puts them in the smooth to transitional regime owing to the low atmospheric density).

(4) "Log-scale" Terminology: The use of the "log-scale" terminology seems unjustified given that ratios of actual (non-log) GS values are reported everywhere in the text. A log-scale ratio difference of 2.75 would imply that coarse grains are between two and three orders of magnitude coarser than fine grains, which isn't the case per Fig. 3. I may be missing something, in which case the text should be clarified; otherwise, I would recommend removing any mention of "log-scale" in the text. If the authors do mean something else by log width and opt to maintain this terminology, it should be

formally introduced (besides being mentioned in the introduction) in the topic sentence of the "Max-Ratio" paragraph.

(5) Application to Remote Sensing: I do not understand why remote sensing was selected as the prime application for the paper's results, given that grain size distributions seem to be required to apply the model, and that even grain size distributions acquired in situ by rovers on Mars (Weitz et al., 2018) appear to be non-sufficient to apply the model. Determining grain size from remote sensing is an extremely difficult inverse problem to start with, and none of the existing orbiting spacecraft would be able to discriminate between crests and bulk materials (e.g., CRISM has a spatial resolution of 10 m/pix and VSWIR is largely insensitive to grain size, e.g., Lapotre et al., 2017; TES and THEMIS have spatial resolutions of 100 m/pix, and even in the TIR determining grain size is challenging, e.g., Edwards et al., 2018).

Edwards, Christopher S., et al. "The thermophysical properties of the Bagnold Dunes, Mars: Ground-truthing orbital data." *Journal of Geophysical Research: Planets* 123.5 (2018): 1307-1326.

Lapotre, Mathieu GA, et al. "Compositional variations in sands of the Bagnold Dunes, Gale Crater, Mars, from visible-shortwave infrared spectroscopy and comparison with ground truth from the Curiosity rover." *Journal of Geophysical Research: Planets* 122.12 (2017): 2489-2509.

(6) Figure 1: The transport regime subplot in Figure 1 is redundant with Figure 2. I do not believe that Nature Communications publishes graphical abstracts, so if this figure will be part of the main text, it shouldn't duplicate a subsequent figure. It is also very confusing as to why the axes are not orthogonal to each other and how this plot meshes with the underlying transport frequency until the reader gets to the part on how the GSD relates to transport regime. I made several suggestions to streamline the flow of figures in the attached PDF.

(7) Tone & Textual Edits: The manuscript's tone is highly conversational, mixing subjective commentary with science results. The manuscript also contains numerous awkward phrasings or grammatical mistakes. I provided suggestions for textual edits, which although they are minor when taken individually, amount to significant rewriting.

On the basis of this review, I would recommend the paper to be sent back to the authors for major revisions before it could be considered for publication in Nature Communications.

Detailed Comments.

Main text: see attached PDF.

Supplementary Materials: p. S2: "Galileo number"

Reviewer #3:

Remarks to the Author:

The authors present a framework for understanding how the sediment transport regime of aeolian megaripples relates to their grain size distribution. While the general concepts introduced are known and understood to most workers in the discipline, the presentation of these concepts in a cohesive framework (notably in Table 1 & Figure 2) represents an important advance. In addition to providing this simple consolidated framework, the authors have collected field data from their own previous work and that of others. While the comparability of different sampling methodologies for this approach is questionable, this is not the fault of the authors and they have addressed these limitations. Rather than necessarily being a flaw, the lack of standardization in field sampling practices could perhaps be more emphasized by the authors as a limitation on their own capacity to use field data for model validation. The work is presented in enough detail to be reproduced and to be tested on other field

data. This work represents an important scientific contribution to a type of geomorphic feature that remains relatively poorly understood. I believe this work to be of significant interest to those in the field of aeolian geomorphology, and of more general interest because of the ongoing discussion of aeolian features on other planets. I have provided suggestions for improvement and points of interest/discussion below.

Lines 42-46 Yes, this point is of particular importance - it is not necessarily that 'megaripples are bimodal', but rather 'samples of megaripple stoss slopes taken with some depth are typically bimodal'.

Lines 53-60. The intention of this pair of sentences, and particularly the words 'fragile' and 'mature', may be unclear to readers. In this context, I understand fragility to mean the susceptibility of a megaripple to being flattened/destroyed. This fragility is a product of the thickness of the armor layer (which is itself a product of time and the available sediment and the volatility of the wind regime). I'm not certain that 'gusts that exceed the prevailing wind strength' fully captures the notion of 'fragility'. Perhaps the coarse-grain saltation threshold in the notation of this paper is a more effective term to use here in providing a quantitative term associated with 'fragility'. With regards to 'maturity', the term is being used to refer to megaripples developing a capacity to be 'anti-fragile' or 'resilient' (i.e., a thicker layer of coarse material). The terminology here is convoluted, because in some cases readers may understand 'maturity' to be synonymous with 'stable', which is not necessarily the case. I'm not sure there is a perfect way to introduce these concepts and avoid confusion for all readers, but perhaps the most intuitive way to sequence the introduction of these concepts is:

1. Description of how megaripples 'mature'
 2. Description of how this maturation process is distinct from that of dunes and ripples
 3. Description of how the maturity of megaripples relates to their fragility
- Because the findings of the paper relate so directly to these terms, I feel they should be more clearly established.

Lines 70-81 This, to my reading, connects very closely to the 'impact fractionation' concept of Anderson and Bunas (1993, Nature). I'd invite the authors to consider reviewing this work and including a reference to it in the context of this discussion, if they agree that it is appropriate.

Lines 81-92 Perhaps a brief explanation of Lämmel et al. 2018's 'reptation dune' concept would be of use to the reader here. In the granulometric and transport mode sense, ripples can be thought of as being 'characterized' by saltation events (perhaps more precisely, by the capacity of virtually all grains to saltate). In megaripples, this saltation capacity is limited to some fine fraction of the GSD. Therefore, reptation becomes the 'characteristic transport mode' of megaripples. This mechanistic difference is essential because it takes multiple reptation/creep events for a grain to reach the crest. This dependence on multiple transport events for cross-brink transport, as opposed to a single saltation event to pass over a typical ripple, is then used to justify the terminology of megaripples as 'reptation dunes'. Where some might disagree is that the dividing line between ripples and dunes should be restricted to features where saltation/reptation mechanisms explain the scaling (ripples) and aerodynamic mechanisms explain the scaling (dunes). The two are of course connected by the notion of the 'saturation transient/length', so the difference may be pedantic to some. Of course, completely explaining this debate is well beyond the scope of this work! Nevertheless, I'd invite the authors to consider these points and consider whether the value of introducing the notion of 'mini-dunes' in the context of this paper is worth the confusion it may cause to some readers. If they feel it is necessary, I am satisfied with the description as it is currently written.

Additionally, I believe the contribution of the work of Jerolmack et al. (2006) is being understated; while not as quantitatively rigorous as the modelling presented here, that paper does describe and provide most of the same threshold-based framework for explaining megaripple development (Figure 5 and Section 3.4. of that paper) as is presented in this work. I'd invite the authors to consider giving more emphasis on the value provided by that work.

Lines 100-104 Yes, this is correct. Near the end of Chapter 3 of Bagnold (1941) (page 35 in my copy), he notes "a high speed grain in saltation can by impact move a surface grain six times its diameter, or more than 200 times its own weight". Much of the interpretation of this relies on how the reader decides to understand 'move' - in your conception of 'move', reptation requires enough energy to move over a neighbouring grain. Perhaps Bagnold's use of 'move' refers to movements of even less than 1 grain diameter (i.e., what some might refer to strictly as 'creep')? This of course depends on 'pocket geometries' and other factors. Regardless, this is interesting discussion.

Line 133 I believe 'rooted' is what is intended here.

Figure II The notion of unimodal transport with large values of $d(c)/d(f)$ is an interesting case - one would not expect highly prominent megaripples (i.e., low ripple index), but might expect some flatter 'megaripples', with clear sorting of coarser grains at the crest because the nature of the saltation/reptation regime in a bidisperse grain size distribution would still be distinct from that in a unimodal regime... and so the features produced with still have megaripple-type characteristics. This is not a criticism, but this figure prompts a very interesting thought exercise in 'mapping' bedform morphologies onto this phase diagram.

Line 235 I believe 'Figure III' is the intended reference here. Additionally, there is no 'bold green line' in the figure.

Lines 235-237 Yes - excellent point. One could hypothesize that for a particularly volatile/variable wind regime, the GSD bimodality is weakened because of grains serving as both saltating and reptating grains depending on the wind. Coupling a wind regime to these GSDs is a natural next step, but beyond the scope of this work.

Line 246 Again, I think 'Figure III' is the intended reference.

Lines 282-306 Excellent discussion of modelling limitations in the context of the complex reality of variable wind speeds. This is such important discussion, thank you for including it.

Lines 373-376 Yes - this is based on the assumption that all grains present are or were once capable of reptation. Some megaripples likely exist in an erosional environment where reptation (in the hopping-over-neighbouring-grain sense used here) is no longer a very active process because a source of finer grains has been winnowed (i.e, a formerly reptating population has now become creep-limited or immobile, and megaripples have stagnated).

Line 610 'been'

We would like to thank all reviewers for their very helpful suggestions. We believe that their constructive comments considerably strengthened the paper. All modifications in the manuscript are highlighted in blue (stylistic edits) or red (edited contents).

Response to Reviewer #1

My opinion about this paper: Great science, thanks for doing this great work. However, I believe that the next time you should try writing a little more to the point, to facilitate the readers' life.

We thank the reviewer for the appreciation of our work and the helpful suggestions for further improvements.

1. *1 Following this overall assessment, my major criticism is that you should acknowledge others' successful megaripple models. If I remember right, the first successful simulation of megaripples was published by Anderson and Bunas (1993), which is Ref. 31 in Lämmel et al. (2018) (i.e., Ref. 31 of your Ref. 31). As you know, they implemented the idea of megaripple formation and inverse grading due to differential hop lengths of fine/coarse reptating particles, and their model inspired other successful models as well, such as Makse (2000), not cited in your paper, but cited in Lämmel et al. (2018) as Ref. 32. Moreover, I think Yizhaq (2004) is also inspired by Anderson and Bunas (1993) to some extent, but with the difference that in Yizhaq (2004) (as well as in Lämmel et al. (2018) and in your paper) the fine grains saltate and the coarse grains reptate. The reader should be informed what your work is adding to the model of Anderson and Bunas (1993), and your paper should acknowledge this preceding work.*

In the revision, we now acknowledge previous studies that are based on the distinction between saltating and reptating grains, including the pioneering study by Anderson and Bunas and others (see lines 60 - 77). These models predict a spatial sorting assuming bidisperse sand (discrete GSDs). In contrast, we derive a theory applicable to continuum GSDs and predict how they evolve in the context of megaripple evolution. Importantly, our theory does not make or require specific assumptions about the underlying instability mechanism leading to megaripple formation. Rather, its conceptualization requires only two ingredients that are not controversial (see lines 112 - 120). We tried to better clarify the novelties of our study throughout the text.

2. *2 Building on my comment above, the curious reader might raise the question whether it is possible to reconcile Fig. 5 of your paper with the great model of Anderson and Bunas (1993). After all, you don't need a (strict) bimodal sand model for Fig. 5 to be valid, right? It would be a great add to the manuscript if you explicitly address this question with a discussion. I think you already do so at different parts of the manuscript, but doing so more explicitly by*

referring to Anderson and Bunas (1993) would be pertinent owing to the historical relevance of their work.

Figure 5 tests our model with data from field sites with continuous and very broad GSDs. In contrast, the model by Anderson and Bunas (and most other models) assumes a strict bidisperse GSDs as a precondition. While spatial sorting occurs also in this earlier model, it does not change the ratio between the two prescribed grain sizes. In particular, Anderson & Bunas did not make predictions about the emergence of a bimodal GSD from a continuum GSD. We clarified this in the revised introduction.

3. *3 Line 174 - 179: Do you mean that the only new result is Fig. 5 then?*

No. We now explain the novelties of our study very carefully in the introduction and highlight them also throughout the text.

4. *4 Fig. 3: The labels on the y-axis are missing, "Frequency (%)" - where are the values of the frequency?*

Done. Thanks.

5. *5 Line 470 - 474: I fully agree. However, I can wholeheartedly recommend the authors to add - if possible - somewhere in the paper an alternative version of Eq. (1) where the most critical parameters, V_s , $\mu_r C_z$, appear as a function of the nondimensional numbers (Galileo, Reynolds, Stokes), and physical parameters/constants (gravity, dielectric constant, whatsoever), so that the field/planetary geomorphologist will be able to more easily put all numbers in the calculator and use your Nat Comm paper in remote sensing missions. This is a recommendation that is not critical for acceptance, so if you see that there is no way to do a good job here (which would be unfortunate), then you can disregard this comment.*

This implicit formula is the best we could achieve without compromising too much of its accuracy. Inspired by your comment, we inserted the typical regimes for Earth and Mars in Fig. 4, such that a one can more easily extract approximate values.

6. *A bonus comment to round up the discussion: Is this work somehow contributing relevant knowledge to understand the origin(s) of the transverse megaripple instability?*

No, it is not because describing transverse megaripple instability would require to specify explicitly the symmetry-breaking mechanism underlying the megaripple instability. One of the major advantages of our present work is that it is insensitive to such details.

Response to Reviewer #2

Summary. Megaripples are eolian bedforms found in windblown deserts of Earth and Mars. They are characterized by polydisperse grain-size distributions, with coarser grains concentrated near the bedform crests, and that typically display longer wavelengths the smaller impact ripples. In their manuscript, Tholen et al. develop a model to explain the grain size distribution (GSD) at the crests of mature megaripples through bimodal transport, with finer grains moving in saltation and coarser grains moving in reptation. A novel result from the model is that the width of the coarse-grain mode in the GSD of megaripple crests encodes information about formative winds in a predictive fashion, which according to the authors may have applications in remote-sensing studies of both terrestrial and extraterrestrial surfaces.

Overall Evaluation. The manuscript reports on an interesting model prediction which could possibly have applications as a (paleo)environmental indicator. The model formulation is skillful, and figures are of overall high quality. However, I noted several important weaknesses in the manuscript that, in my opinion, would preclude the manuscript from publication in *Nature Communications* in its current form:

We are glad the reviewer appreciates our study.

1. (1) *Novelty:* It is not immediately clear how the paper’s framework is different from the conceptual model of Lämmel et al. (2018). Figure 2 directly maps onto the latter model: “a window of wind strengths can be identified, in which the coarse grains do not themselves saltate but merely creep or ‘reptate’ over the sand bed, slowly driven by the saltating fraction of fine grains. This wind range is delimited by the threshold wind shear stresses τ_{ct} and τ_{cr} for saltation and reptation of the coarse grains and thereby sensitive to the ratio a_c/a_f of the characteristic coarse and fine sand grain diameters a_c and a_f ” (direct quote from Lämmel et al., 2018). As it stands, the conceptual model appears to be merely the same as that of Lämmel et al., using a different parameterization of transport modes, and with one added prediction (that of the max width). To claim novelty and warrant publication in *Nature Communications*, I would strongly encourage the authors to clarify and emphasize the distinction between Lämmel et al. (2018) and this manuscript, and the specific contributions of this study.

There are major conceptual and modeling differences between Lämmel et al. (2018) and our study. They seem mainly important for claiming novelty rather than for an actual reader of the article, which is why we did not dwell on them very much. We are now more explicit in the revised version.

Conceptual differences

- In contrast to Lämmel et al. (2018), we explicitly take into account that the coarsest

grains are immobile (lines 265 - 282). Only this very ingredient allows us to predict the max-size ratio, whereas Lämmel et al. (2018) only predicted the existence of a minimum between the fine-grain and coarse-grain peaks.

- Our model is based on an improved conceptual understanding of the saltation threshold, which was published in Pächtz et al. (2020 & 2021). We consider the threshold as an entrainment-independent “rebound” threshold (see discussion in lines 583 - 595), whereas Lämmel et al. (2018) assumes that entrainment by splash plays a crucial role for the saltation threshold.
- We generalized the conceptual model for the saltation threshold of Pächtz et al. (2021) to the situation of fine grains saltating along a bed of coarse grains.

Modeling differences

- In contrast to Lämmel et al. (2018), we modeled the fine- and coarse-grain saltation thresholds, which is the very reason we were able to predict the max-size ratio.
- With our quantitative model, we are able to predict all thresholds for arbitrary aeolian environmental conditions (now more emphasized in lines 190 - 191), whereas Lämmel et al. (2018) relied on heuristic estimate for terrestrial conditions.
- To estimate the reptation threshold, Lämmel et al. (2018) modeled a grain-bed collision via a semi-empirical effective restitution coefficient, which they left unspecified. We calculated this coefficient based on an idealized model for an optimum grain-bed collision.

2. *(2) Model Description: The methods section is very helpful; however, the manuscript as it stands does not provide the reader with any information on what physics went into producing the results in the main text. I would strongly recommend that the authors include a bit more information about the “ingredients” that went into the model in the main text. This does not need to be a detailed discussion as the Methods does a great job at that – just enough information for the reader to have some idea of what went into the model without having to dig into the full mathematical formulation.*

We fully agree, in the main part, it was not clearly written. In lines 202 - 224, we try to go a bit more into the details. We hope that one can now grasp more intuitively the ingredients and concepts of the underlying model without going through the whole method section.

3. *(3) Parameterization of Wind Profile: The wind profile is parameterized using a z_0 formulation that is appropriate for aerodynamically rough beds with no active transport. However, unless I am missing something, the modeled wind profile is intended to be representative of*

wind events that transport the fine fraction in saltation, and should thus be a function of the height of the saltation layer rather than grain roughness. The authors should at the very least justify their use of an aerodynamically rough/no transport formulation of z_0 . Note also that martian winds, when no active transport occurs, are not so clearly aerodynamically rough (roughness-scale Reynolds number puts them in the smooth to transitional regime owing to the low atmospheric density).

Although fine grains are transported in saltation, their effect on the wind profile is neglected because the fine-grain mass flux is highly undersaturated due to the armoring coarse-grain layer, which limits the supply of fine grains into the transport layer. For this reason, we can assume a wind profile that is undisturbed by the presence of grain motion. For this undisturbed profile, we use a very general empirical formulation (Eq. M5) that includes not only the log-layer but also the viscous sublayer and buffer layer of the turbulent boundary layer in the case of smooth and transitional conditions (for rough conditions, only the log-layer exists). In this formulation, the nominal zero-velocity elevation z_0 of the log-layer turns from $z_0 = \nu/(9u_*)$ for smooth conditions to $z_0 = d^{(c)}/30$ for rough conditions. Note that the approximation in Eq. 1 is only valid for the fully rough regime, whereas the more complex prediction shown in the inset of Fig. 4 is computed using the general velocity profile. However, Eq. 1 has a wide range of applicability, since the approximate description in terms of a rough regime is usually appropriate because of the large size of coarse grains at the bed surface.

4. (4) *“Log-scale” Terminology: The use of the “log-scale” terminology seems unjustified given that ratios of actual (non-log) GS values are reported everywhere in the text. A log-scale ratio difference of 2.75 would imply that coarse grains are between two and three orders of magnitude coarser than fine grains, which isn’t the case per Fig. 3. I may be missing something, in which case the text should be clarified; otherwise, I would recommend removing any mention of “log-scale” in the text. If the authors do mean something else by log width and opt to maintain this terminology, it should be formally introduced (besides being mentioned in the introduction) in the topic sentence of the “Max-Ratio” paragraph.*

We now introduce the term log-scale in the introduction and explicitly define it in Sec. 2c. The ratio of the (non-log) GS values that identify the boundaries of the reptation regime (i.e., the max-size ratio) is by definition directly related to the width of this regime, if it is plotted on a logarithmic scale. Namely, the log-scale width is defined as

$$\ln(\max(d^{(c)})) - \ln(\max(d^{(f)})) = \ln(\max(d^{(c)})/\max(d^{(f)})) .$$

5. (5) *Application to Remote Sensing: I do not understand why remote sensing was selected as the prime application for the paper’s results, given that grain size distributions seem to be required to apply the model, and that even grain size distributions acquired in situ by rovers on Mars (Weitz et al., 2018) appear to be non-sufficient to apply the model. Determining grain size from remote sensing is an extremely difficult inverse problem to start with, and*

none of the existing orbiting spacecraft would be able to discriminate between crests and bulk materials (e.g., CRISM has a spatial resolution of 10 m/pix and VSWIR is largely insensitive to grain size, e.g., Lapotre et al., 2017; TES and THEMIS have spatial resolutions of 100 m/pix, and even in the TIR determining grain size is challenging, e.g., Edwards et al., 2018).

We thank the reviewer for detecting an important error we made. We agree, the word remote sensing is not correct in this context (we corrected it). However, although probably extremely expensive, the grain size distributions acquired in situ by rovers on Mars (Weitz et al., 2018 and already Jerolmack et al., 2006) are actually sufficient to apply the model as the resolution is high enough. In fact, we used the GSD of Jerolmack et al., 2006, obtained from images recorded by the Opportunity rover. In contrast to Jerolmack et al., 2006, the data by Weitz et al., 2018 exhibit too low counts of grains in the coarse-grain peak such that it was impossible to extract the peak width in a reliable way.

We selected the debate about martian mid-size bedforms as a prime application because it is currently controversial how to discriminate between all these bedforms that are similar in their morphology. The strength of our theory is that the quantitative (and robust) prediction of the coarse grain peak width could be used as a strong indicator for classifying a bedform as a megaripple. In contrast, the mere observation of a bimodal GSD could in principle be explained by other segregation processes (or by a bimodal sand source). Note that spatial grain sorting and bedform evolution even occurs for bidisperse sands when both grain sizes are saltating. This may suggest that not all bedforms with an armouring layer on the crest should be addressed as megaripples in the strict sense of our model description. We clarified this in lines 501 - 516.

6. *(6) Figure 1: The transport regime subplot in Figure 1 is redundant with Figure 2. I do not believe that Nature Communications publishes graphical abstracts, so if this figure will be part of the main text, it shouldn't duplicate a subsequent figure. It is also very confusing as to why the axes are not orthogonal to each other and how this plot meshes with the underlying transport frequency until the reader gets to the part on how the GSD relates to transport regime. I made several suggestions to streamline the flow of figures in the attached PDF.*

Although we are firmly convinced that this inset is very instructive as to how Figs. 2 & 3 are mutually related, we decided to shift it into the Supplementary information (Fig. S1) and replaced it by a new figure that reveals the weak correlation between the modes, which were classically used to characterize the GSDs.

7. *Tone & Textual Edits: The manuscript's tone is highly conversational, mixing subjective commentary with science results. The manuscript also contains numerous awkward phrasings or grammatical mistakes. I provided suggestions for textual edits, which although they are minor when taken individually, amount to significant rewriting.*

Detailed Comments.

Main text: see attached PDF.

Thank you very much for all these comments, which are highly appreciated. We adopted the large majority of them in the revised version, which was thereby significantly improved. Please, find our arguments for the few exception below.

- *Line 7: What is meant by volatile here? "Volatile" in the geosciences is most often used to refer to a substance that easily evaporates, which makes its use confusing here. Please clarify.*

In the revision, we provide a definition of this term to avoid potential misunderstandings.

- *Lines 25 - 33:*

We have decided to keep our language in the first paragraph of the introduction because Nature Communications is a journal with an interdisciplinary readership and encourages the authors to carefully introduce technical terms.

- *Fig. 2 would be more impactful if it was framed in terms of morphodynamics (i.e., inverted with table I - the figure would be about megaripple dynamics and the table would explain how it relates to transport). Or the figure could even report both morphodynamic and transport modes in each regime (e.g., "unimodal transport: megaripple destruction," "bimodal transport: megaripple formation," etc.).*

We try to arrange Tab. 1 and Fig. 2 side by side in the final manuscript as they are inseparable (the table acts as a kind of legend to the figure). However, we had to make concessions in their designs. For example, if we were to add morphodynamical and/or sorting modes of megaripples to the figure, as the reviewer suggests, it would be overloaded. Likewise, with morphodynamical modes only, it would be misleading as the phase diagram is calculated based on the transport modes. However, we are open to further suggestions.

- *Line 153: Does this refer to the fluid or impact threshold of coarse grains? Does it differ from what it would be if the bed was monodisperse and made of the coarse population only? Notably, if this refers to the impact threshold, how is it modified by the presence of impacting finer grains?*

It refers to the impact threshold. Here, the impact threshold is just called saltation threshold and in the model defined as the smallest Shields number for which a periodic grain trajectory exists. This is conceptually different from the classical modeling of the impact threshold via quantifying the ejection of grains by impacts. We decided to not mention these (and other) details in the main text as it would probably lead to confusion (they are mentioned in Sec. M1.B). They are thoroughly discussed in

two recent papers (Pächtz et al., 2020 & 2021). Furthermore, there are two saltation thresholds, one for the fine grains bouncing along a bed of coarse grains and one for coarse grains bouncing along a bed of coarse grains. This means the bed is always considered to be made of only coarse grains. The dependency of the fine-grain saltation threshold on the size ratio between bed and saltating grains is non-trivial and shown in Figure M1.

- *Line 214: This header is ambiguous because "bimodal" qualifies a general distribution and is not uniquely tied to transport. For example., "bimodal sand" is often used as a synonym for "bidisperse sand." I'd suggest something like "From bidisperse sand to bimodal sand transport"*

We partly agree, it is a bit ambiguous. We changed it to "From bidisperse to continuum bimodal sands". However, we did not follow the reviewer's suggestion as we explicitly want to refer to the continuum grain size distribution.

- *It is highly non-intuitive that a more complex system reduces the phenomenological complexity; this sentence needs to be unpacked - please step the reader through the reasoning here. I think that this true because $\max(d_f) = \min(d_c)$?*

The observation $\max(d_f) = \min(d_c)$ is true, but it is not the crucial point in this argumentation. The crucial point is that, even though a natural polydisperse GSD is "more complex" than a bidisperse GSD, the complexity of the corresponding polydisperse transport is reduced because the system can adapt to the prevailing wind strength due to a continuous range of grain sizes potentially available for saltation. This additional freedom of choice leads to the collapse of the transport phase diagram. We clarify this in lines 245 - 252.

- *Line 243: It would be helpful to refer to Figure 3 here for visual support of where $\max(d^c)$ and $\max(d^f)$ are in a natural GSD.*

We would prefer to not refer to Fig. 3 at this very point as it explains only the collapse of the transport phase diagram in the case of a continuum GSD, which only requires the additional degree of freedom just mentioned above. Note that, directly after this sentence, we refer to Fig. 3.

- *z_0 is not the roughness height, but the distance from the boundary at which the velocity goes to zero. It is only a function of roughness height under aerodynamically rough conditions and no active transport. Unless I am missing something, this is an incorrect assumption in this case and z_0 should be taken as a function of the thickness of the saltation cloud.*

We clarify in lines 358 - 360 that z_0 is the nominal zero-level of the undisturbed log-layer wind profile (which is appropriate for the considered undersaturated conditions associated with megaripple evolution) for aerodynamically rough conditions. Furthermore, for the modeling of the wind profile, see reply to the comment 3.

8. Supplementary Materials: p. S2: “Galileo number”

Done. Thanks.

Response to Reviewer #3

1. *The authors present a framework for understanding how the sediment transport regime of aeolian megaripples relates to their grain size distribution. While the general concepts introduced are known and understood to most workers in the discipline, the presentation of these concepts in a cohesive framework (notably in Table 1 & Figure 2) represents an important advance. In addition to providing this simple consolidated framework, the authors have collected field data from their own previous work and that of others. While the comparability of different sampling methodologies for this approach is questionable, this is not the fault of the authors and they have addressed these limitations. Rather than necessarily being a flaw, the lack of standardization in field sampling practices could perhaps be more emphasized by the authors as a limitation on their own capacity to use field data for model validation. The work is presented in enough detail to be reproduced and to be tested on other field data. This work represents an important scientific contribution to a type of geomorphic feature that remains relatively poorly understood. I believe this work to be of significant interest to those in the field of aeolian geomorphology, and of more general interest because of the ongoing discussion of aeolian features on other planets. I have provided suggestions for improvement and points of interest/discussion below.*

We thank the reviewer for her/his appreciation of our work. As the reviewer nicely pointed out, the lack of standardization in field sampling practices limits our capacity to use field data for model validation. We emphasize this issue more strongly in the revised version.

2. *Lines 42-46 Yes, this point is of particular importance - it is not necessarily that 'megaripples are bimodal', but rather 'samples of megaripple stoss slopes taken with some depth are typically bimodal'.*

Yes, thank you, this important difference is emphasized in the introduction together with the mentioned sampling problem.

3. *Lines 53-60. The intention of this pair of sentences, and particularly the words 'fragile' and 'mature', may be unclear to readers. In this context, I understand fragility to mean the susceptibility of a megaripple to being flattened/destroyed. This fragility is a product of the thickness of the armor layer (which is itself a product of time and the available sediment and the volatility of the wind regime). I'm not certain that 'gusts that exceed the prevailing wind strength' fully captures the notion of 'fragility'. Perhaps the coarse-grain saltation threshold in the notation of this paper is a more effective term to use here in providing a quantitative*

term associated with ‘fragility’. With regards to ‘maturity’, the term is being used to refer to megaripples developing a capacity to be ‘anti-fragile’ or ‘resilient’ (i.e., a thicker layer of coarse material). The terminology here is convoluted, because in some cases readers may understand ‘maturity’ to be synonymous with ‘stable’, which is not necessarily the case. I’m not sure there is a perfect way to introduce these concepts and avoid confusion for all readers, but perhaps the most intuitive way to sequence the introduction of these concepts is:

1. Description of how megaripples ‘mature’
2. Description of how this maturation process is distinct from that of dunes and ripples
3. Description of how the maturity of megaripples relates to their fragility

Because the findings of the paper relate so directly to these terms, I feel they should be more clearly established.

We agree, especially the word “mature” was used in different contexts which could lead to confusion. We followed the referee’s advice by rephrasing the term “mature” and introducing the term “fragile” more systematically.

4. *Lines 70-81 This, to my reading, connects very closely to the ‘impact fractionation’ concept of Anderson and Bunas (1993, Nature). I’d invite the authors to consider reviewing this work and including a reference to it in the context of this discussion, if they agree that it is appropriate.*

We now acknowledge previous studies that are based on the distinction between saltating and reptating grains, including the pioneering study by Anderson and Bunas and others (see lines 60 - 86). Note that these models predict a spatial sorting (“impact fractionation”) assuming bidisperse sand in the sense of a discrete GSD as a precondition. In contrast, we derive a theory applicable to continuum GSDs and predict how they evolve in the context of megaripple evolution.

5. *Lines 81-92 Perhaps a brief explanation of Lämmel et al. 2018’s ‘reptation dune’ concept would be of use to the reader here. In the granulometric and transport mode sense, ripples can be thought of as being ‘characterized’ by saltation events (perhaps more precisely, by the capacity of virtually all grains to saltate). In megaripples, this saltation capacity is limited to some fine fraction of the GSD. Therefore, reptation becomes the ‘characteristic transport mode’ of megaripples. This mechanistic difference is essential because it takes multiple reptation/creep events for a grain to reach the crest. This dependence on multiple transport events for cross-brink transport, as opposed to a single saltation event to pass over a typical ripple, is then used to justify the terminology of megaripples as ‘reptation dunes’. Where some might disagree is that the dividing line between ripples and dunes should be restricted to features where saltation/reptation mechanisms explain the scaling (ripples) and aerodynamic mechanisms explain the scaling (dunes). The two are of course connected by the notion of the ‘saturation transient/length’, so the difference may be pedantic to some. Of course, completely explaining this debate is well beyond the scope of this work! Nevertheless,*

I'd invite the authors to consider these points and consider whether the value of introducing the notion of 'mini-dunes' in the context of this paper is worth the confusion it may cause to some readers. If they feel it is necessary, I am satisfied with the description as it is currently written.

These are indeed somewhat subtle issues, and we have taken great care to improve their discussion. A key insight is that our present discussion requires only very little physical input concerning the mechanism of megaripple formation (see lines 112 - 120). It almost exclusively relies on the physics of bimodal sediment transport. We therefore no longer refer to the reptation dune concept.

6. *Additionally, I believe the contribution of the work of Jerolmack et al. (2006) is being understated; while not as quantitatively rigorous as the modelling presented here, that paper does describe and provide most of the same threshold-based framework for explaining megaripple development (Figure 5 and Section 3.4. of that paper) as is presented in this work. I'd invite the authors to consider giving more emphasis on the value provided by that work.*

When introducing the thresholds, we now mention that the distinction between them is not original but can be found in several previous studies (lines 177 - 179).

7. *Lines 100-104 Yes, this is correct. Near the end of Chapter 3 of Bagnold (1941) (page 35 in my copy), he notes "a high speed grain in saltation can by impact move a surface grain six times its diameter, or more than 200 times its own weight". Much of the interpretation of this relies on how the reader decides to understand 'move' - in your conception of 'move', reptation requires enough energy to move over a neighbouring grain. Perhaps Bagnold's use of 'move' refers to movements of even less than 1 grain diameter (i.e., what some might refer to strictly as 'creep')? This of course depends on 'pocket geometries' and other factors. Regardless, this is interesting discussion.*

We are discussing Bagnold's estimation now more detailed throughout the text.

8. *Line 133 I believe 'rooted' is what is intended here.*

Thanks. The sentence is not in the manuscript any more.

9. *Figure II The notion of unimodal transport with large values of $d^{(c)}/d^{(f)}$ is an interesting case - one would not expect highly prominent megaripples (i.e., low ripple index), but might expect some flatter 'megaripples', with clear sorting of coarser grains at the crest because the nature of the saltation/reptation regime in a bidisperse grain size distribution would still be distinct from that in a unimodal regime... and so the features produced with still have megaripple-type characteristics. This is not a criticism, but this figure prompts a very*

interesting thought exercise in 'mapping' bedform morphologies onto this phase diagram.

This is a very good insight, thank you for pointing this out. In fact, it was shown that megaripple-like bedforms also evolve above the coarse-grain saltation threshold. Although these bedforms undergo spatial sorting, their crest GSD probably does not exhibit the same characteristics as “normal” megaripples (with a fixed log-scale width of the coarse grain peak).

10. *Line 235 I believe 'Figure III' is the intended reference here. Additionally, there is no 'bold green line' in the figure.*

Actually, we really mean Fig. 2. At this point, we didn't apply the phase diagram to a measured bimodal GSD as it evolves upon megaripple formation (only in the subsequent paragraph we made this connection). To avoid confusions, we added the bold green line in Fig. 3 and rephrase the explanation of the collapse a bit. We hope that one can now better see the connection explained in the subsequent paragraph.

11. *Lines 235-237 Yes - excellent point. One could hypothesize that for a particularly volatile/variable wind regime, the GSD bimodality is weakened because of grains serving as both saltating and reptating grains depending on the wind. Coupling a wind regime to these GSDs is a natural next step, but beyond the scope of this work.*

Thank you for acknowledging this point. Indeed, we tried to address this problem in lines 295 - 321 (at least in a rough way).

12. *Line 246 Again, I think 'Figure III' is the intended reference.*

Again, here we also mean Figure 2.

13. *Lines 282-306 Excellent discussion of modelling limitations in the context of the complex reality of variable wind speeds. This is such important discussion, thank you for including it.*

Thank you for the appreciation.

14. *Lines 373-376 Yes - this is based on the assumption that all grains present are or were once capable of reptation. Some megaripples likely exist in an erosional environment where reptation (in the hopping-over-neighbouring-grain sense used here) is no longer a very active process because a source of finer grains has been winnowed (i.e, a formerly reptating population has now become creep-limited or immobile, and megaripples have stagnated).*

Thank you for pointing out this particular limiting case. We think that it is related to one of the key points of our study. Without the presence of immobile grains in the sand source, one could not get the defined width of the coarse grain peak.

15. *Line 610 'been'*

Done. Thanks.

Reviewers' Comments:

Reviewer #1:

Remarks to the Author:

The authors revised the paper by taking my comments satisfactorily into account, and I can now recommend publication of this manuscript in Nature Communications.

Reviewer #2:

Remarks to the Author:

Review of the revised manuscript, "Megaripple mechanics: bimodal transport ingrained in bimodal sands"

by Tholen et al. for Nature Communications

Summary. Megaripples are eolian bedforms found in windblown deserts of Earth and Mars. They are characterized by polydisperse grain-size distributions, with coarser grains concentrated near the bedform crests, and that typically display longer wavelengths the smaller impact ripples. In their manuscript, Tholen et al. develop a model to explain the grain size distribution (GSD) at the crests of mature megaripples through bimodal transport, with finer grains moving in saltation and coarser grains moving in reptation. A novel result from the model is that the width of the coarse-grain mode in the GSD of megaripple crests encodes information about formative winds in a predictive fashion, which could have implications for our understanding of terrestrial and extraterrestrial eolian landscapes.

Overall Evaluation. The authors did a great and thorough job at addressing reviews from three different referees, including mine. As per my previous evaluation, the manuscript reports on an interesting model prediction that could have important implications as a (paleo)environmental indicator on Earth and other planets. The model formulation is skillful. Figures are of high quality, and the manuscript's readability and flow greatly improved through revisions. The manuscript will be of interest to a range of planetary and geo-scientists as well as physicists interested in granular mechanics. I have few remaining suggestions – mainly minor edits and one request for a brief, more explicit discussion of potential limitations of the model's applicability due to sediment supply (see below). Based on this evaluation, I recommend publication of the manuscript in Nature Communications pending minor revisions.

Detailed Comments:

(1) Supply limitation: The authors greatly improved their discussion of the controls on the right edge of the coarse-grain peak, and notably, noted that a limited supply of coarse grains that cannot be transported by prevailing winds could lead to deviations from model predictions. This is an extremely important point, as unless the bedforms are forming close to their sediment source or some other process supplies coarse grains locally (e.g., a river), those coarsest grains could have been "filtered" out of transport further upstream along the transport pathway, or worse, they could have never been supplied in the first place (e.g., the parent rock breaks down into fine, easily transportable grains, as could be the case for finely crystalline igneous rocks for example). This point seems particularly critical on Mars, where only winds have transported sand for the past few billions of years, and modern-day erosion primarily occurs through eolian abrasion (and mass wasting to some minor extent). I made a couple of suggestions in the PDF, largely requesting a more intentional discussion of the role of supply limitation.

(2) A few minor edits are suggested in the attached PDF. I do not have any further comments regarding the supplementary materials.

Reviewer #3:

Remarks to the Author:

The authors have made significant changes to the manuscript that have improved its clarity and, to me, signal readiness for publication after a copy edit. All of my significant concerns from the first review have been addressed. Two relatively minor pieces of general feedback and several line edits/comments follow.

General Comments

The use of 'log-scale width' is unusual because it seems to me that this quantity is just a range of grain sizes in units of phi. Unless I've missed something, 'log-scale width' is synonymous with 'phi width'. Since phi is an extremely common and widely understood way to describe grain size, it seems kinder to the reader to just describe this concept using phi. But perhaps I've missed something that makes 'log-scale width' terminology necessary. Since both 'max size ratio' and 'log-scale width' are both simply ways of quantifying the width of the coarse fraction of the GSD, it may be simpler to omit mention of 'log-scale width' to avoid confusing the reader (although I do appreciate that the mention of 'log-scale width' is done to provide another perspective). I leave the decision of terminology to the authors but do strongly suggest simplifying to more common terminology (phi), as it seems that both Reviewer 2 and myself were confused by the current terminology.

In some cases, 'sand' and 'sediment' are used interchangeably. I take 'sand' to refer strictly to sand-sized grains, whereas 'sediment' refers to grains of all sizes. Because some megaripples may include gravel-sized grains, the use of 'sediment' may be more precise in some cases where 'sand' is currently used (e.g., on lines 13 and 26). I recognize that this is pedantic and leave this change to the authors' discretion.

Line comments/edits

Line 28. What is 'size' referring to here? If it refers to wavelength (as it does for megaripples on line 34, then 'decimeter-sized' is likely a better descriptor for impact ripples.

Line 75. 'Troughs' not 'throughs'

Line 182. 'on the basis of' or 'based on' instead of 'on basis'

Lines 433-434. This phrase is very close to that used by Lammel et al 2018. I recognize that several of the authors of this work were also authors of that work, but this phrasing is close enough to be worth quoting directly, citing that work at the end of this sentence, or rephrasing (i.e., the wording was close enough that after reading the sentence it made me think it was a direct quote from Lammel et al 2018).

Line 520. The more recent work of Favaro et al. 2020 (Icarus) may be a more relevant citation than your current citation (62), because this more recent paper includes measurements of grain density from samples rather than anecdotal observation. This is very minor, and I leave this change to the discretion of the authors.

Line 523. 'pressure' not 'pressue'

Line 807. Delete 'from'.

Figure S4-S8 x axis units appear as if they should be μm , not mm.

Supplementary Material Table S3 caption: Measurements, not 'measurments'.

Fig S9 Caption: troughs, not 'throughs'

Response to Reviewer 2

1. Review of the revised manuscript, “Megaripple mechanics: bimodal transport ingrained in bimodal sands” by Tholen et al. for *Nature Communications*

Summary. Megaripples are eolian bedforms found in windblown deserts of Earth and Mars. They are characterized by polydisperse grain-size distributions, with coarser grains concentrated near the bedform crests, and that typically display longer wavelengths the smaller impact ripples. In their manuscript, Tholen et al. develop a model to explain the grain size distribution (GSD) at the crests of mature megaripples through bimodal transport, with finer grains moving in saltation and coarser grains moving in reptation. A novel result from the model is that the width of the coarse-grain mode in the GSD of megaripple crests encodes information about formative winds in a predictive fashion, which could have implications for our understanding of terrestrial and extraterrestrial eolian landscapes.

Overall Evaluation. The authors did a great and thorough job at addressing reviews from three different referees, including mine. As per my previous evaluation, the manuscript reports on an interesting model prediction that could have important implications as a (paleo)environmental indicator on Earth and other planets. The model formulation is skillful. Figures are of high quality, and the manuscript’s readability and flow greatly improved through revisions. The manuscript will be of interest to a range of planetary and geo-scientists as well as physicists interested in granular mechanics. I have few remaining suggestions – mainly minor edits and one request for a brief, more explicit discussion of potential limitations of the model’s applicability due to sediment supply (see below). Based on this evaluation, I recommend publication of the manuscript in *Nature Communications* pending minor revisions.

Detailed Comments: (1) *Supply limitation:* The authors greatly improved their discussion of the controls on the right edge of the coarse-grain peak, and notably, noted that a limited supply of coarse grains that cannot be transported by prevailing winds could lead to deviations from model predictions. This is an extremely important point, as unless the bedforms are forming close to their sediment source or some other process supplies coarse grains locally (e.g., a river), those coarsest grains could have been “filtered out” of transport further upstream along the transport pathway, or worse, they could have never been supplied in the first place (e.g., the parent rock breaks down into fine, easily transportable grains, as could be the case for finely crystalline igneous rocks for example). This point seems particularly critical on Mars, where only winds have transported sand for the past few billions of years, and modern-day erosion primarily occurs through eolian abrasion (and mass wasting to some minor extent). I made a couple of suggestions in the PDF, largely requesting a more intentional discussion of the role of supply limitation.

In the attached PDF, the reviewer further specified what s/he means with this comment: As we explained in the paper, there are two possible reasons for why a bedform's coarse-grain peak deviates from the model prediction: (i) coarse grains have been "filtered out" or (ii) the bedform is not a megaripple. The reviewer asks how do we assess which of these two reasons is the right one? Similarly, s/he suggests that on Mars it may be the standard case that coarse grains have been filtered out.

Our response: First, if coarse grains have been filtered out, then the width of the coarse-grain peak would be smaller than the prediction, but not larger. That is, if one were to find a bedform with a grain size distribution that exhibits a larger-than-predicted coarse-grain peak, it would imply that this bedform is not a megaripple. This is now clarified in the paper. Second, we note that the two available measurements of grain size distributions on Mars' surface we are aware of (Jerolmack et al. 2006 and Weitz et al. 2018) both contain very coarse grains. This in itself suggests that the occurrence of very coarse grains on Martian soil may not be as rare as the reviewer here suggests.

2. (2) A few minor edits are suggested in the attached PDF. I do not have any further comments regarding the supplementary materials.

We followed most of the Reviewer's minor suggestions. In line 111 we decided to keep the word "theory" rather than the referee's suggestion "model". Nevertheless, we followed the referee's advice and took care about the difference between the terms "theory" and "model" throughout the manuscript by keeping the interdisciplinary readership in mind.

Response to Reviewer 3

1. The authors have made significant changes to the manuscript that have improved its clarity and, to me, signal readiness for publication after a copy edit. All of my significant concerns from the first review have been addressed. Two relatively minor pieces of general feedback and several line edits/comments follow.

General Comments

The use of 'log-scale width' is unusual because it seems to me that this quantity is just a range of grain sizes in units of ϕ . Unless I've missed something, 'log-scale width' is synonymous with 'phi width'. Since ϕ is an extremely common and widely understood way to describe grain size, it seems kinder to the reader to just describe this concept using ϕ . But perhaps I've missed something that makes 'log-scale width' terminology necessary. Since both 'max size ratio' and 'log-scale width' are both simply ways of quantifying the width of the coarse

fraction of the GSD, it may be simpler to omit mention of ‘log-scale width’ to avoid confusing the reader (although I do appreciate that the mention of ‘log-scale width’ is done to provide another perspective). I leave the decision of terminology to the authors but do strongly suggest simplifying to more common terminology (ϕ), as it seems that both Reviewer 2 and myself were confused by the current terminology.

We now clarify in the abstract and the first time we mention ‘log-scale width’ in the main text that this width is proportional to the Krumbein ϕ scale width.

2. *In some cases, ‘sand’ and ‘sediment’ are used interchangeably. I take ‘sand’ to refer strictly to sand-sized grains, whereas ‘sediment’ refers to grains of all sizes. Because some megaripples may include gravel-sized grains, the use of ‘sediment’ may be more precise in some cases where ‘sand’ is currently used (e.g., on lines 13 and 26). I recognize that this is pedantic and leave this change to the authors’ discretion.*

As the reviewer pointed out, only material up to 2mm is considered sand in Geology and Soil sciences. However, the more relevant field for us is Geomorphology, with a strong focus on soil erosion/particle transport by water and wind. Basically, all the eroded/transported material can be defined using the general term “sediment”, no matter the size or minerology. However, we are looking for a more specific term that excludes the suspended transport of aeolian clay and silt ($d < 60\mu m$). As such, we use the term “sand” because it refers to grain sizes larger than $60\mu m$ and, in the context of dune and ripple formation processes, it is commonly not limited to grain sizes smaller than 2mm. Looking at it the other way around, “aeolian sediment ripple” may not reflect well our case because it can be considered also as clay-size ripple. In the revised manuscript, we now note that we define sand as sediment of grains larger than about $60\mu m$ (lines 31-32).

3. *Line comments/edits*

Line 28. What is ‘size’ referring to here? If it refers to wavelength (as it does for megaripples on line 34, then ‘decimeter-sized’ is likely a better descriptor for impact ripples.

We changed to ‘decimeter-sized’.

4. *Line 75. ‘Troughs’ not ‘throughs’*

Corrected.

5. *Line 182. ‘on the basis of’ or ‘based on’ instead of ‘on basis’*

We now state ‘on the basis of’.

6. *Lines 433-434. This phrase is very close to that used by Lammel et al 2018. I recognize that several of the authors of this work were also authors of that work, but this phrasing is close enough to be worth quoting directly, citing that work at the end of this sentence, or rephrasing (i.e., the wording was close enough that after reading the sentence it made me think it was a direct quote from Lammel et al 2018).*

We now cite Lämmel et al. here.

7. *Line 520. The more recent work of Favaro et al. 2020 (Icarus) may be a more relevant citation than your current citation (62), because this more recent paper includes measurements of grain density from samples rather than anecdotal observation. This is very minor, and I leave this change to the discretion of the authors.*

Favaro et al. (2020) is now being cited here.

8. *Line 523. ‘pressure’ not ‘pressue’*

Corrected.

9. *Line 807. Delete ‘from’.*

Done.

10. *Figure S4-S8 x axis units appear as if they should be μm , not mm.*

Corrected.

11. *Supplementary Material Table S3 caption: Measurements, not ‘measurments’.*

Corrected.

12. *Fig S9 Caption: troughs, not ‘throughs’*

Corrected.